



**Observational Study on the Variability of Mixed Layer Depth in the Bering**
**Sea and the Chukchi Sea in the Summer of 2019**
**Xiaohui Jiao[1], Jicai Zhang[1,*], Chunyan Li[2]**
[1] Institute of Physical Oceanography, Ocean College, Zhejiang University, Zhoushan
316021, China.
[2] Department of Oceanography and Coastal Sciences, College of the Coast and
Environment, Louisiana State University, Baton Rouge 70803, U.S.A.
Corresponding author: Jicai Zhang (jicai_zhang@163.com)
**Key Points:**
- The mixed layer depth in the Bering Sea Slope was greater than that in the
Bering Sea Basin due to the circulation and eddy in the slope
- The mixed layer depth was influenced by the Bering Sea Anadyr Water and the
Alaska Coastal Water in the Bering Sea shelf
- The further northward, the more important role the salinity played in
determining the mixed layer depth in the Chukchi Sea.



## Abstract

Based on the high-resolution CTD data from 58 stations in the Bering Sea and the Chukchi Sea in the summer of 2019, the mixed layer depth (MLD) was obtained according to the density difference threshold method. It was verified that the MLD could be estimated more accurately by using a criterion of $0.125$ kg$/m^3$ in this region. The MLD in the Bering Sea basin was larger than that in the Bering Sea shelf, and both of them were smaller than that in the Bering Sea slope. The MLD increased northward both in the Chukchi Sea shelf and the Chukchi Sea slope. The farther northward, the greater the difference between the MLD calculated from temperature (MLDt) and the MLD calculated from density (MLDd) was, and the more important the role of salinity was in determining the MLD. The larger MLD (refer to MLDd specifically) in the Bering Sea slope might be due to the enhancement of mixing caused by the Bering Slope Current (BSC) and eddies. The horizontal advection of the Bering Sea Anadyr Water and the Alaska Coastal Water in the Bering Sea shelf led to the shallower MLD in the central transition zone. The northward increase of the MLD in the Chukchi Sea might be related to the low-salinity seawater resulting from the melting of sea ice in summer. The spatial variation of MLD was more closely related to the surface momentum flux than the sea surface buoyancy flux, and the wave had little effect.



## 1 Introduction

The dynamics in the Bering Sea and the Chukchi Sea have an important impact on global climate change (Hu et al., 2010). The mean sea level in the Pacific is higher than that of the Arctic Ocean (Coachman & Aagaard, 1966). Therefore, the average flow in the Bering Strait is northward (Overland et al., 1996), and the average annual flow is about 0.83 Sv (Roach et al., 1995). The net northward transport has a marked impact on Arctic sea ice, as it feeds a subsurface temperature maximum under the ice-pack in winter (Woodgate et al., 2010; Woodgate et al., 2012). It also influences the global hydrologic cycle (Serreze et al., 2006), and the global thermohaline circulation (Shaffer & Bendtsen, 1994; Wadley & Bigg, 2002). The seasonality of the northward heat and freshwater transport is strongly influenced by the mixed layer (Woodgate, 2018).

The exploration of the upper ocean mixed layer depth (MLD hereafter) will be very beneficial for the study of the northward heat and water transport through the Bering Strait and its climatic effects. Water mass formation and circulation (Hanawa & Talley, 2001; Stommel, 1979), sea-air exchange (Frankignoul & Hasselmann, 1977; Kraus & Turner, 1967; Rodgers et al., 2014; Stevenson & Niiler, 1983), biogeochemical processes (Fasham et al., 1990; Fauchereau et al., 2011; Sverdrup, 1953), transfer of heat between ocean and sea ice (Sirevaag, et al., 2011), and melting/freezing of sea ice (Polyakov et al., 2013) are highly sensitive to the variations of MLD. The mixed layer affects the process of biological production by importing or exporting phytoplankton and nutrients into the light-transmitting layer (Chen et al., 1994; Ohlmann et al., 1996). The accuracy of MLD also plays an important role in the reliability of climate model results (Belcher et al., 2012). Therefore, the exact information of MLD is vital for the study of physical oceanography, climate change, and relative subjects.

Many factors can affect the variations of MLD, such as surface heat and freshwater fluxes, horizontal advection, wind stress (Hu & Wang, 2010), Langmuir circulation (Li et al., 2013), sea ice (Peralta-Ferriz & Woodgate, 2015), eddy (Gaube et al., 2019), and the oceanographic structure including temperature and salinity below the mixed layer



(Hanawa & Toba, 1981; Tully, 1964). The strengthening or weakening of stratification
caused by the air-sea kinetic energy exchange or buoyancy flux in the surface of the
ocean will also change the MLD (Deardorff et al., 1969; Kato & Phillips, 1969; Kraus
& Turner, 1967; Large et al., 1994; McWilliams et al., 1997; McWilliams et al., 2009;
Price et al., 1986). Under the effect of wind, waves, and Langmuir circulation, the MLD
become deeper, which has been proved by many researches based on theory,
observations, and numerical models (Bruneau & Toumi, 2016; Li et al., 2013; Wu et al.,

71  2015).

The variation of the mixed layer in this region is complicated due to the influence of the
circulation, eddy, wind, and sea ice. The major current in the Bering Sea is a cyclonic
circulation with the Bering Slope Current (BSC hereafter) along the Bering Sea slope,
the Kamchatka Current in the northwest, the Alaskan Coastal Current in the northeast,
and the Aleutian North Slope Current in the north of Aleutian Islands (Panteleev et al.,
2012). A sketch of them is shown in Figure 1 (Danielson et al., 2014). The hydrological
characteristics in the Bering Sea are influenced by the Pacific Ocean due to the water
exchange between the Bering Sea and the Pacific Ocean with the major inflow through
the Near Strait and outflow through Kamchatka Strait (Stabeno & Reed, 1994). Strong
eddy activities have been observed along the Bering Slope due to the instability of the
BSC (Mizobata et al., 2006; Okkonen, 2001). The seasonal variation of wind in this
region is remarkable. In winter, the Aleutian Low moves southward, and most areas of
the Bering Sea are controlled by polar cold air (Rodionov et al., 2007). Northwest wind
prevails and part of the sea surface will be frozen (Zhang et al., 2010). In summer, south
wind prevails, and all the sea ice is melted (Dong et al., 2019; Serreze et al., 2016). As a
result, the MLD varies drastically in time and space. Since the Bering Sea is separated
into the coastal shelf, middle shelf, outer shelf, and basin by fronts approximately
overlying the 50m isobaths, 100 isobaths, and continental slope (Coachman and
Charnell, 1979; Kachel et al., 2002; Kinder and Coachman, 1978; Schumacher et al.,
1979), the vertical structure of the water column and physical process can be different,
including the mixed layer. The upper layer is wind-mixed and the bottom layer is





tidally-mixed in the middle shelf of the Bering Sea (Schumacher and Stabenow, 1998).
The sea ice in the Arctic showed a trend of later freeze up and a trend toward earlier
melt onset (Markus et al., 2009), which might change the temporal variation of the
mixing layer. The MLD can be less than 20 m in summer while reaching more than 500
m in winter in subpolar latitudes (Monterey, 1997). Seasonal variability of the MLD is
associated with heat flux in the Bering Sea (Kara et al., 2000).

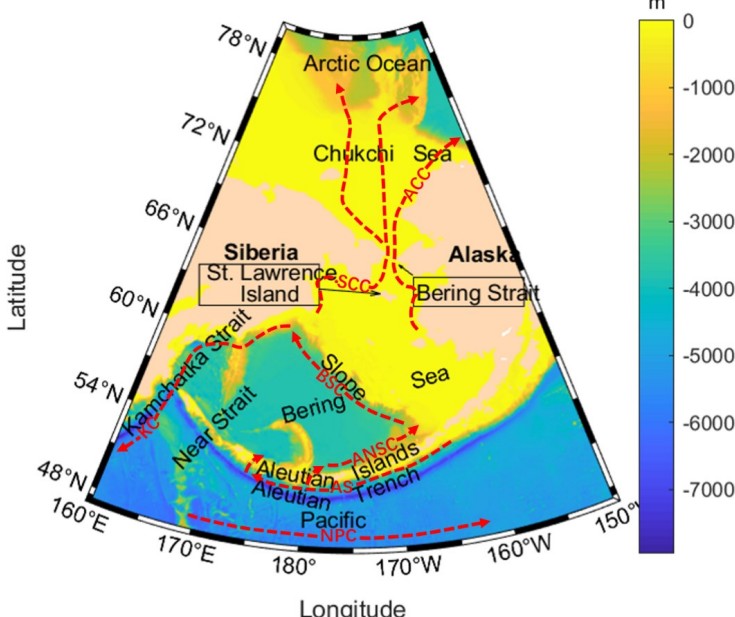


**Figure 1.** Topography, bathymetry, and circulation in the Bering Sea, Chukchi Sea,
and adjacent region. Abbreviations include: ACC = Alaskan Coastal Current; SCC =
Siberian Coastal Current; KC = Kamchatka Current; BSC = Bering Slope Current;
ANSC = Aleutian North Slope Current; AS = Alaskan Stream; NPC = North Pacific
Current. (Danielson et al., 2014; Kawaguchi & Nishioka, 2020)
Thanks to the rapid growth of Argo observations in the past decade, the MLD in most
of the global ocean has been better studied (Holte et al., 2017). There are several global
MLD datasets available (Carton et al., 2008; de Boyer Montégut et al., 2004; Hosoda et
al., 2010; Monterey, 1997; Schmidtko et al., 2013). Some of them use the temperature
to get the MLD without the salinity data. Besides, most of these previous works mainly
focused on the global ocean, but less on the Bering Sea and the Chukchi Sea, except for
the study on the MLD in the Arctic   (Peralta-Ferriz & Woodgate, 2015). It was found




that the overall mixed layer gradually became shallower from 1978 to 2012 by studying
the seasonal and inter-annual changes of the MLD in the Arctic region (Peralta-Ferriz
& Woodgate, 2015), which covered the Chukchi Sea but lacked the data in the Bering
Sea. At present, due to the lack of high spatial resolution datasets and further
understanding of the ocean dynamics, the global model simulation results still have
large deviations (Belcher et al., 2012; D'Asaro, 2014; DuVivier et al., 2018). To
evaluate and reduce the model deficiency in modeling the values of MLD, direct
observations of MLD are of great significance, especially for the Arctic ocean. In this
paper, the field observational data sampled during the summer of 2019 will be analyzed
to study the spatial variations of MLD in the Bering Sea and the Chukchi Sea, which
will benefit the model calibration and evaluation, air-sea interaction, and climate
change, etc.
This paper is organized as follows. Section 2 introduces the data and methods. The
analyzing results are given in Sect. 3. The processes related to the spatial variation of
MLD are discussed in Sect. 4. Section 5 presents the summary and conclusions.
**2 Data and methods**
**2.1. Study area**
The Bering Strait connecting the Bering Sea and the Chukchi Sea has a depth of about
50 m and a width of 80 km and is the only direct ocean gateway for the exchange of
matter and energy between the Arctic Ocean and the Pacific Ocean (Woodgate et al.,
2005). The north-south span of the Bering Sea is around 1500 km and the east-west
span is more than 2300 km (Figure 2). The northeast part of the Bering Sea is a shallow
continental shelf, the depth of which ranges from tens of meters to about two hundred
meters (Dong et al., 2019). The southwest part is a basin with a depth of more than 3000
m in most areas. A sharp change of the water depth from 200 m on the northeastern side
to 2000 m on the southwestern side exists within less than 100 km in width between the
Bering Sea basin and the Bering Sea shelf (Figure 2). The southern part of the Chukchi





Sea is a continental shelf with a depth of about 50 m (Woodgate et al., 2005), and the
northern part is a basin with a depth of more than 2000 m.
**2.2. Data**
The in-situ observational data used in this paper were obtained during the 10th Chinese
National Arctic Research Expedition from Aug. 10 to Sept. 30, 2019. As shown in
Figure 2, 58 stations were distributed in BL, BR, BS, R, BT, and M transection (Figure
2). The BL section was located across the western Bering Sea basin, continental slope,
and continental shelf. The BS section was located in the northern Bering Sea shelf. The
BR section was located in the eastern Bering Sea slope and shelf. The R section was
located in the southern Chukchi Sea shelf, and BT was located in the northern Chukchi
Sea shelf. M section was located in the Chukchi Sea slope. These sections are
representatives of this region.
Two kinds of observations were carried out during the expedition: transect
hydrographic investigations and ship-borne ADCP observation. LADCP/CTD
operations were performed to get profiles of temperature, salinity, and velocity at 58
stations (Figure 2 and Table 3) during the transect hydrographic investigations. The
SBE 911 Plus (Table 1), Teledyne RDI WHMariner 300kHz, and Teledyne RDI OS
38kHz (Table 2) were used in the transect hydrographic investigations (stations in
Figure 2). Ship-borne ADCP measurement was carried out while the ship was in
motion to get the current profile of the upper ocean along the track. The surface
temperature and salinity measurements were made as well in the underway
observations. The SeaBird FerryBox (Table 1) and Teledyne RDI WHSentinel 300kHz
(Table 2) were used in the underway observations.
The Lowered ADCP used to observe the current velocity was a 300 kHz Teledyne RDI
WHN Workhorse Sentinel. During the observation, the descending rate of the
instrument was controlled to 1.0 $m/s$. The signal frequency of lowered ADCP used to
observe the ocean current was 300 kHz, and its max range was 110 m. The real
sampling depth changes with the particles, temperature, and salinity of the seawater.



The vertical sampling resolution of the Lowered ADCP was 2 ~ 8 m, and the sampling
frequency was 1 Hz. The Lamont-Doherty Earth Observatory (LDEO) software based
on the inverse method (Visbeck, 2002) was used to process the data from the Lowered
ADCP.
The temperature, conductivity, pressure, and dissolved oxygen were measured in the
transect hydrographic investigations. The lowered CTD measuring the temperature and
salinity was SBE-911-Plus direct-reading temperature-conductivity-depth profiler
(Table 1), and the sampling frequency was 24 Hz. Since a second redundant pair of
temperature, conductivity, pressure sensors, and the pump were installed, we could
perform better quality control on multiple parameters.
The signal frequency of ship-based underway ADCP was 38 kHz in the basin and the
continental slope and was 300 kHz in the continental shelf (Table 2). Since the depth in
the continental shelf was shallow, the bottom tracking was used for better accuracy in
the ocean current velocity measurements. In the deeper ocean basin and the continental
slope, the velocity of the vessel calculated by GPS was used to correct the current
velocity. The ship-based underway equipment for the surface temperature and salinity
measurement was the SeaBird FerryBox.





**Table 1.** Equipment for temperature and salinity measurement.

| Instrument | Model | Sampling frequency | Conductivity resolution | Temperature resolution (°C) | Pressure resolution (db) |
|---|---|---|---|---|---|
| Lowered CTD | SBE 911 Plus | 24 | 0.00004 | 0.0002 | 0.001 |
| Underway multi-element system | SeaBird FerryBox | 1 | 0.005 | 0.0001 | -- |


**Table 2.** ADCP Model

| Instrument | Model | Bin size | Sampling depth | No. Bins | Pings/Ens | Time/Ping(s) |
|---|---|---|---|---|---|---|
| Lowered ADCP | Teledyne RDI WHSentinel 300kHz | 2~8m | 110m | 14~50 | 1 | 1 |
| Underway ADCP1 | Teledyne RDI WHMariner 300kHz | 4m | 110m | 50 | 1 | 0.5 |
| Underway ADCP2 | Teledyne RDI OS 38kHz | 24m | 960m | 40 | 1 | 3 |





**Table 3.** The longitude, latitude, and sampling start time of the 58 stations.

| Station | Longitude | Latitude | Date and time | Station | Longitude | Latitude | Date and time |
|---------|-----------|----------|---------------|---------|-----------|----------|---------------|
| BL01 | 171.87E | 54.58N | 24/08 06:33:22 | BS04 | 170.13W | 64.33N | 29/08 08:20:48 |
| BL02 | 172.77E | 55.27N | 24/08 13:02:20 | BS05 | 169.41W | 64.33N | 29/08 10:11:39 |
| BL03 | 174.57E | 56.57N | 24/08 23:36:43 | BS06 | 168.71W | 64.33N | 29/08 12:05:17 |
| BL04 | 175.60E | 57.39N | 25/08 07:27:26 | BS07 | 168.11W | 64.33N | 29/08 14:01:38 |
| BL05 | 177.41E | 58.30N | 25/08 18:03:27 | BS08 | 167.45W | 64.37N | 29/08 15:14:42 |
| BL06 | 178.41E | 58.72N | 27/08 00:06:14 | BT12 | 167.12W | 74.32N | 03/09 05:54:45 |
| BL07 | 179.51W | 60.04N | 27/08 13:46:35 | BT13 | 167.82W | 74.75N | 01/09 08:00:24 |
| BL08 | 179.00W | 60.40N | 27/08 17:28:25 | BT14 | 167.85W | 75.03N | 01/09 11:17:22 |
| BL09 | 178.21W | 60.80N | 27/08 21:38:45 | BT15 | 167.82W | 75.33N | 01/09 14:36:37 |
| BL10 | 177.23W | 61.29N | 28/08 02:32:42 | BT16 | 167.80W | 75.64N | 01/09 18:11:55 |
| BL11 | 176.17W | 61.93N | 28/08 07:25:10 | BT25 | 167.81W | 74.74N | 02/09 20:59:52 |
| BL12 | 175.01W | 62.59N | 28/08 12:18:21 | BT26 | 171.21W | 74.60N | 01/09 04:20:21 |
| BL13 | 173.43W | 63.29N | 28/08 18:19:56 | BT27 | 169.32W | 74.35N | 03/09 09:42:37 |
| BL14 | 172.40W | 63.77N | 28/08 22:08:49 | M11 | 166.44W | 74.80N | 02/09 18:08:27 |
| BR00 | 174.09W | 56.95N | 08/09 15:52:39 | M12 | 172.00W | 75.21N | 02/09 14:48:17 |
| BR01 | 173.69W | 57.41N | 08/09 11:26:24 | M13 | 172.01W | 75.61N | 02/09 10:55:12 |
| BR02 | 173.22W | 57.90N | 08/09 07:31:37 | M14 | 172.00W | 76.03N | 02/09 02:53:16 |
| BR03 | 172.73W | 58.40N | 08/09 04:22:07 | M15 | 171.96W | 75.82N | 01/09 22:43:37 |
| BR04 | 172.25W | 58.91N | 08/09 00:26:27 | R01 | 169.87W | 66.21N | 30/08 02:09:33 |
| BR05 | 171.30W | 59.90N | 07/09 17:25:13 | R02 | 168.75W | 66.89N | 30/08 05:40:07 |
| BR06 | 170.35W | 60.91N | 07/09 11:09:28 | R03 | 168.75W | 67.50N | 30/08 09:16:57 |
| BR07 | 169.67W | 61.65N | 07/09 06:18:06 | R04 | 168.75W | 68.19N | 30/08 13:09:21 |
| BR08 | 168.89W | 62.40N | 07/09 01:13:13 | R05 | 168.76W | 68.81N | 30/08 17:17:20 |
| BR09 | 168.42W | 62.91N | 06/09 21:16:20 | R06 | 168.75W | 69.53N | 30/08 21:11:26 |
| BR10 | 167.93W | 63.40N | 06/09 18:11:50 | R07 | 168.75W | 70.33N | 31/08 02:08:10 |
| BR11 | 167.47W | 63.90N | 06/09 14:01:53 | R08 | 168.75W | 71.17N | 31/08 07:18:38 |
| BS01 | 171.39W | 64.32N | 29/08 02:52:43 | R09 | 168.75W | 71.99N | 31/08 11:56:35 |
| BS02 | 170.82W | 64.33N | 29/08 04:42:27 | R10 | 168.74W | 72.90N | 31/08 16:38:02 |
| BS03 | 170.12W | 64.33N | 29/08 06:31:27 | R11 | 168.74W | 74.15N | 31/08 23:57:32 |



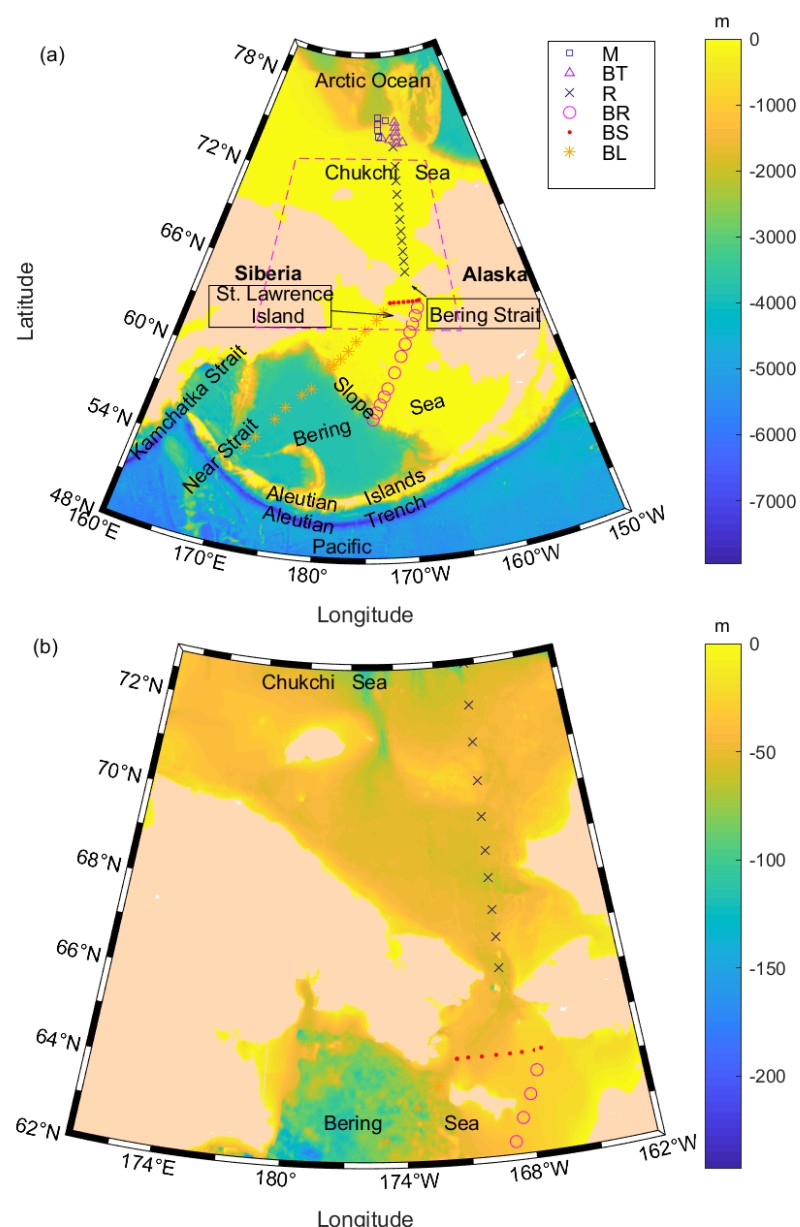


**Figure 2.** (a) showed the distribution of the 58 observation stations. The asterisks, dots, circles, crosses, triangle, and squares represented the BL, BS, BR, R, BT, and M transection, respectively. (b) showed the bathymetry and topography in the dashed line rectangle in (a).


193 The automatic meteorological station installed at a height of 16 m above the sea level

194 was used to measure wind speed, air temperature, air pressure, and humidity throughout

195 the cruise. The speed of the ship estimated by GPS was used to calculate wind speed.

196 The sampling interval is 1 minute.

197 The CCMP reanalysis wind data at a height of 10 m above the sea level was also used in

198 the present work. The spatial resolution of CCMP data is 0.25°, and the temporal

199 resolution is 6 hours (Wentz et al, 2015). The sea surface heat flux and water flux were

200 obtained from the CFSv2 (Saha et al., 2011). The Bering Sea level was obtained from

201 the combined measurements of several altimeters (available online at

202 https://resources.marine.copernicus.eu/?option=com_csw&view=details&product_id=

203 SEALEVEL_GLO_PHY_L4_NRT_OBSERVATIONS_008_046). The significant

204 wave height was obtained from the COPERNICUS MARINE ENVIRONMENT

205 MONITORING SERVICE (available online at

206 https://resources.marine.copernicus.eu/?option=com_csw&view=details&product_id=

207 WAVE_GLO_WAV_L4_SWH_NRT_OBSERVATIONS_014_003). The

208 bathymetric data used in this paper was from ETOPO1 (Amante & Eakins, 2009).

209 **2.3. The criterion for the MLD**

210 In practice, several ocean parameters, including temperature, salinity, density, turbulent

211 mixing, and dissolved oxygen, can be used to calculate the value of MLD. Methods to

212 estimate MLD include difference threshold (de Boyer Montégut et al., 2004), gradient

213 threshold (Lukas & Lindstrom, 1991), curvature method (Lorbacher et al., 2006), split

214 and merge method (Thomson & Fine, 2003), etc. For example, the potential density

215 difference threshold method was used to calculate the MLD with a criterion of 0.01 kg

216 m$^{-3}$ and a reference depth of 10 m (Smyth et al., 1996; Wijesekera & Gregg, 1996). The

217 potential density gradient threshold method determined the MLD as a depth range

218 where the vertical gradient of the potential density was below a critical value, and many

219 researchers used a gradient threshold of 0.1 $kg/m^4$ (Lukas & Lindstrom, 1991). The

220 MLD calculated by the least-squares regression and integration method represented the



depth of the thermocline to a greater extent. Some researchers proposed a
split-and-merge method, which could be used not only to calculate the MLD but also to
describe other marine vertical structural features (Thomson & Fine, 2003). Therefore,
the difference threshold and gradient threshold are better choices. Although dissolved
oxygen could be used to calculate MLD, the results are not accurate where Ekman
pumping is strong or marine productivity is active. Besides, due to the existence of the
vertical density-compensated layer, the MLD estimated by density maybe not accurate
in winter (de Boyer Montégut et al., 2004).
In this paper, the most widely adopted difference threshold method was used to
estimate the MLD. The criterion was critical as the small vertical fluctuations of the
temperature (red rectangle in Figure 3 (b)) and density within the mixed layer might
bring confusion when calculating the MLD. The temperature profiles of all stations
could be divided into three classes. BR01 was a station of type A, where the
temperature of the mixed layer was almost completely homogeneous, and the
temperature gradient was very small. BR00 was a station of type B, where the
temperature of the mixed layer had local extremum. As a result, if a small threshold was
used, the calculated MLD would be shallower than the real MLD. BL08 was a station of
type C, and the temperature of the mixed layer changed significantly vertically. In
general, the temperature had an apparent trend of decreasing with depth, and small
amplitude fluctuations were imposed on this trend.


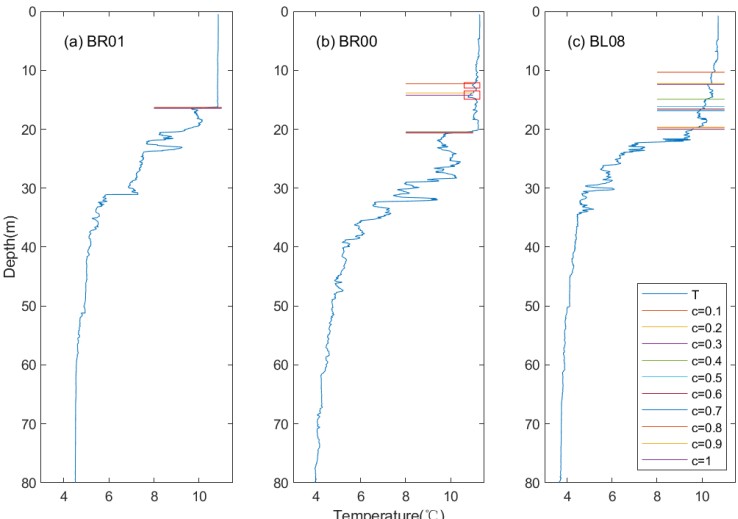

**Figure 3.** Three types of temperature profiles. Horizontal lines in different colors showed different MLDs responding to a group of temperature criteria. (a) showed the type A temperature profile, which had almost the same MLD using different criteria. (b) showed the type B temperature profile, and the MLD calculated from this temperature profile using different temperature criteria was distributed around the local extremum. (c) showed the type C temperature profile; the MLD calculated from type C temperature profile using different temperature criteria had more difference, and the distributions were more dispersed. The variable c in the legend represented the temperature criteria which ranged from 0.1 to 1 °C.

As Figure 3 and Figure 4 showed, the MLD could be divided into three types as well. The type A stations had almost the same MLD using different criteria. The MLD calculated from type B stations using different temperature criteria was distributed around the local extremum. The MLD calculated from type C stations using different temperature criteria had more difference, and the distributions were more dispersed.

A criterion that could overcome the influence of the local extremum was used in the Bering Sea and the Chukchi Sea. The suitable criterion for the temperature difference method was 0.5 $°C$. To explore the role of temperature and salinity in the spatial variation of the MLD, the MLDd (MLD from the density) and the MLDt (MLD from the temperature) were both calculated, and in the following, MLD specifically refers to MLDd. The density profiles at some stations also had local extremum, and the same



principle was followed in determining the criterion for the density difference method.
The MLDd was defined as the depth at which density differed from that of the depth of
5 m by 0.125 $\mathrm{kg}/m^3$. This is consistent with the previous research. Considering the
diurnal variation of MLD, the criteria may be 0.125 $\mathrm{kg}/m^3$ in climate research (Kara
et al., 2000).

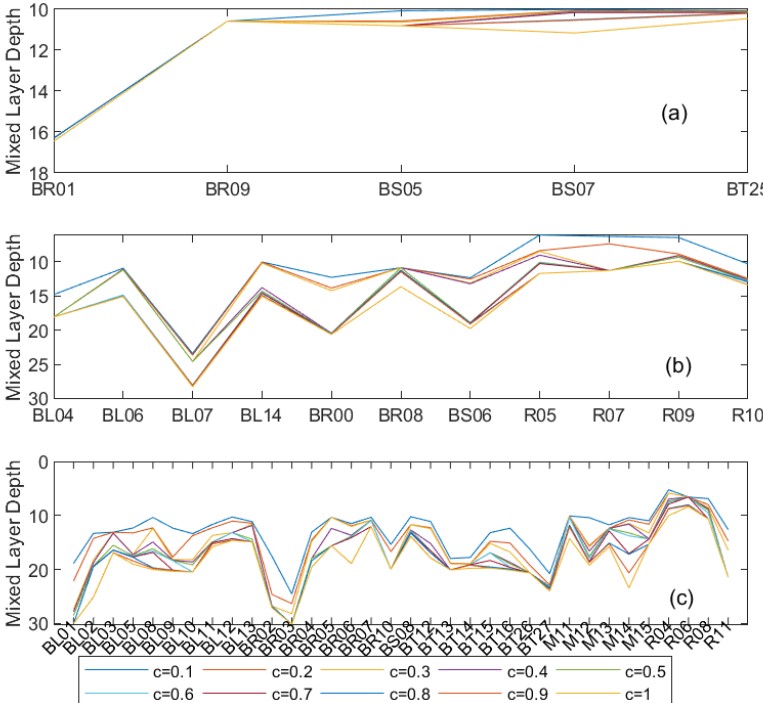


**Figure 4.** The sensitivity of the MLDt at different stations to criteria is different. (a)
Type A: different stations had almost the same MLDt for different criteria; (b) Type B:
stations had several MLD around the local extremum calculated by using different
criteria; (c) Type C: The MLDt calculated from different criteria changed uniformly
with the criteria. The variable c in the legend represented the temperature criteria
which ranged from 0.1 to 1 ℃.
**2.4. Stratification index**
The MLD is constrained by the stratification of the ocean. To explore the role of the
temperature and the salinity in the stratification, a stratification index was calculated. It



is defined as the potential energy of a water column relative to the completely mixed
state (Ladd & Stabeno, 2012):
$$V = -\int_{-h}^{0} \left( \rho - \langle \rho \rangle \right) g z dz \qquad (1)$$

$$\langle \rho \rangle = \frac{1}{h} \int_{-h}^{0} \rho dz \qquad (2)$$

where $\rho$ is the seawater density, and $h$ is the depth of the water column. For a
vertically completely mixed water column, $V=0$. The warmer and fresher the ocean
surface is, the greater $V$ is. Wind stress and negative buoyancy flux may strengthen
vertical mixing (Prasad, 2004), and make $V$ smaller. Changes in the salinity and
temperature both lead to variations of stratification. Their contributions to stratification
were explored. For the stratification index due to temperature (salinity), the density
profile $\rho$ is calculated using the temperature (salinity) profile and a vertically averaged
salinity (temperature).
**3. Result analysis**
**3.1. The salinity and temperature**
The spatial variation of MLD is closely related to hydrography and ocean dynamics.
The temperature and salinity characteristics between the Bering Sea basin and the
continental shelf were significantly different (Figure 5 and Figure 6). The temperature
and salinity profiles showed that the seawater in the upper ocean in the Bering Sea
Basin was warmer and saltier than that in the Bering Sea Shelf (Figure 5). The sea
surface temperature and salinity had a similar pattern (Figure 6). The upper ocean
above 30 m in the Bering Sea basin had the characteristics of high temperature and low
salinity. The sea surface temperature was about 11.5 ℃ and the sea surface salinity was
about 33 in the Bering Sea basin, while they were 10.5 ℃ and 32.2 respectively in the
Bering Sea shelf. There was a cold water mass with a depth range of 50-200m and a
core temperature slightly lower than 3 ℃ in the middle layer. The temperature of the
bottom cold water mass in the southern continental shelf was similar to that of the



middle cold water mass in the basin, but the bottom cold water mass was shallower due
to terrain constraints on the shelf. The temperature of the bottom cold water mass
decreased from 3 ℃ in the south to 1 ℃ in the north.
The hydrographic features between the northeast and northwest of the continental shelf
were different. There was a cold water mass with a core temperature of 2 ℃ in the west,
and the salinity was higher than 32.5. In the east, the temperature was higher than 9 ℃
and the salinity was significantly lower than that in the west. In the east, the density of
high-temperature and low-salinity water was smaller, which had the characteristics of
the Alaska Coastal Water. It might be affected by the Yukon River's freshwater. The
density of low-temperature and high-salinity water on the west side was larger and had
the characteristics of the Anadyr Water.
There were high-temperature and low-salinity water masses with the temperature range
of 1~10 ℃ and salinity of 28~30 in the upper layer of the Chukchi Sea shelf and the
continental slope (Figure 8). The sea surface temperature gradually decreased from
10 ℃ in the south to 2 ℃ in the north and the sea surface salinity decreased from 30
to 28 from south to north. There was a low-temperature and high-salinity water mass on
the bottom. The temperature of the bottom water decreased from 4 to -1.3 ℃ from
south to north, while the salinity also decreased from 32 to 30 (Figure 8). There was a
middle cold water mass with a core temperature of -1.8 ℃ in the depth range of 40m ~
150m below the surface warm water in the Chukchi Sea slope.
**3.2. Characteristics of MLD in the Bering Sea**
The spatial distributions of MLD between the basin located in the southwest of the
Bering Sea and the continental shelf located in the northeast had significantly different
characteristics.
The BL section was representative due to its wide span of space. The stations BL02 -
BL06 were located in the Bering Sea basin, and the MLD at these stations were all
greater than 15 m (Figure 7 (a)). The maximum value of the MLD in the Bering Sea
basin reached 18.72 m, and it was observed at the BL02 station located in the southern



Bering Sea basin. The MLDd and MLDt had little difference at stations BL02 - BL05,
but the difference between the MLDd and the MLDt approached 4 m at BL06 station
(Figure 7 (d)). In contrast, the MLD observed at stations BL11 - BL14 which were
located in the west of the Bering Sea shelf was shallower than 15 m. And the minimum
of the MLD at the BL section was 6.23 m, which was observed at BL14 station. The
BL14 station was located in the northwestern Bering Sea shelf. It should be noted that
the MLDt (14.30 m) is much larger than MLDd (6.25 m) at BL14 station. The
maximum of the MLD at the BL section was 30.03 m. It occurred at the BL01 station,
which was located in the continental slope on the north of the Aleutian Island. The
MLD at BL07 station located in the Bering Sea slope was 25.32 m. Both of the MLD at
BL01 and BL07 stations were markedly larger than those MLD in the Bering Sea basin
and the Bering Sea shelf.
The stations BR00 - BR03 were on the Bering Sea slope. The maximum MLD reached
30.17 m, and it occurred at BR03 station (Figure 7 (b)). The minimum MLD was 16.32
m at these stations and almost larger than all the MLD in the Bering Sea shelf, including
the stations BR04 - BR10. The average MLD at stations BR04 - BR09 was 13.72 m,
much smaller than the 23.44 m in the Bering Sea slope. The stations BR10 and BR11
were in the northeast of the Bering Sea shelf, and it can be seen that the isohaline and
the isothermal were almost vertical there (Figure 5 (b) and (e)). Corresponding to that,
the MLD at BR10 and BR11 stations were dramatically greater than those at other BR
stations in the Bering Sea shelf.
The western BS section was under the influence of the water mass named Anadyr
Water in the northwestern Bering Sea, and the eastern BS section was under the
influence of the Alaska Coastal Water in the northeastern Bering Sea. Thus the BS
section represented the MLD under the influence of the advection of these two water
masses. As the water column at BS01 - BS03 was well-mixed, the MLD were all larger
than 35 m. On the contrary, the MLDt was zero there. The MLD in the northeast was
much smaller and had a value of 6 m - 13 m. As the denser Anadyr Water intruded
eastward under the Alaska Coastal Water (Figure 5 (c)), the MLD in the transition zone,


including stations BS04 and BS05, was smaller (Figure 7 (c)). It should be noted that
the MLDt was significantly greater than MLDd in the northeastern Bering Sea shelf.
Overall, the MLD in the Bering Sea basin was greater than those on the continental
shelf. The MLD in the continental slope of the Bering Sea was significantly greater than
those in the basin and the continental shelf. The MLD was larger than 30 m as the water
column was well-mixed in the northwestern Bering Sea shelf, but the MLD decreased
toward the east. The MLDd was generally larger than MLDt in the Bering Sea basin,
the Bering Sea slope, and the southern Bering Sea shelf. But in the northern Bering Sea
shelf, the MLDt was significantly larger than MLDd.



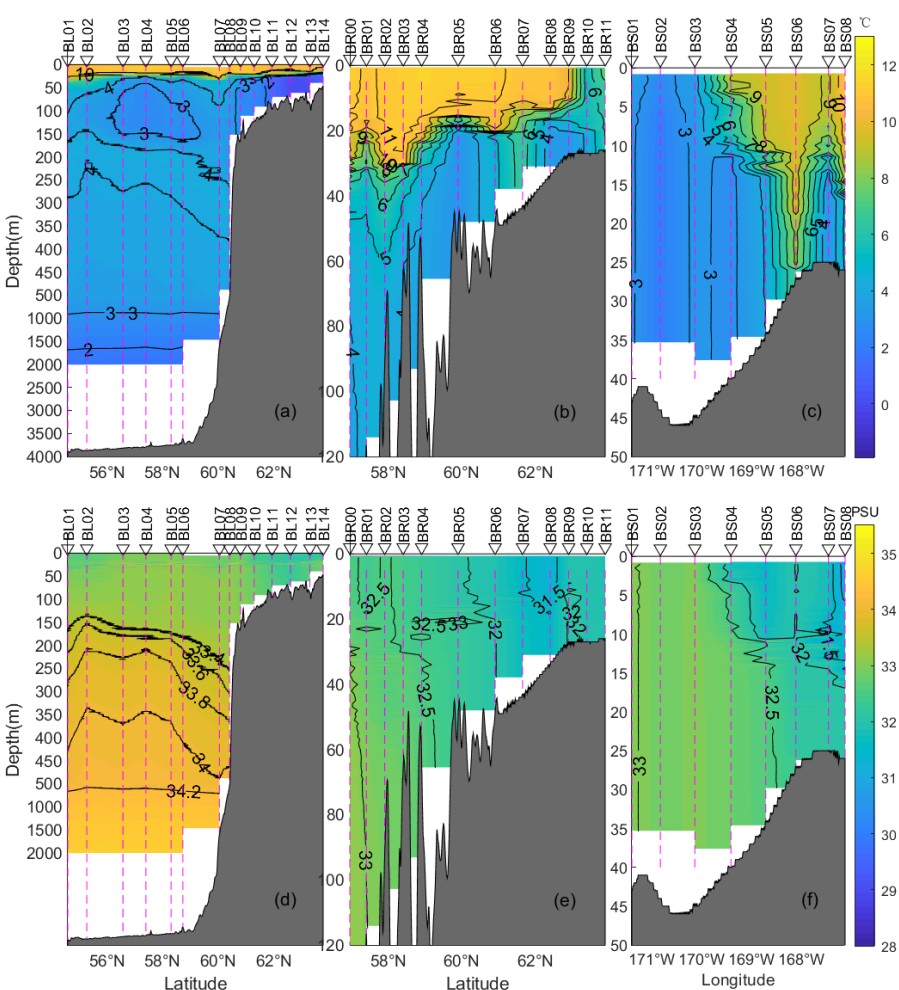

**Figure 5** The upper panels and the lower panels represent the temperature and salinity profiles, respectively. The left (a, d), middle (b, e), and right (c, f) column represent the section of BL, BR, and BS, respectively.

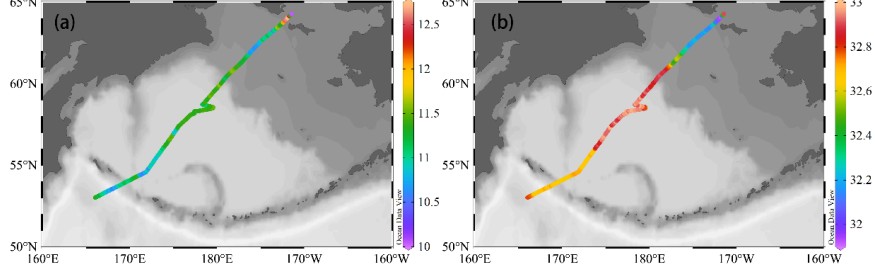

**Figure 6.** The sea surface temperature (a) and salinity (b) in the Bering Sea.



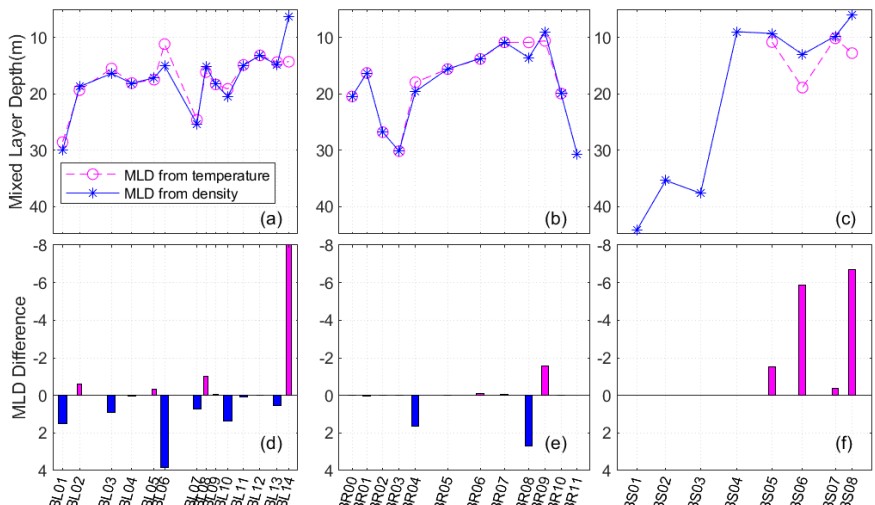

**Figure 7.** The upper panel represents the MLD from temperature and density. The lower panel represents the difference between MLDd and MLDt. The left (a, d), middle (b, e), and right (c, f) column represent the section of BL, BR, and BS respectively. The magenta dash lines represent the MLD calculated from the temperature, and the blue solid lines represent the MLD calculated from the density. The magenta bar means that the MLDt is larger than the MLDd, and the blue bar means that the MLDd is larger than the MLDt. Notice that the Y-axis is reversed.

**3.3. Characteristics of MLD in the Chukchi Sea**

The Chukchi Sea is on the north of the Bering Strait. The depth of the continental shelf in the south is about 50 m and it spans several hundreds of kilometers. The MLD at the R section was in the range of 4-12 m. In general, the MLD increased at a rate of $4.5 \times 10^{-6}$ northward along the R section (Figure 9 (a)).

The northward increase was also observed in the BT section. The maximum MLD was 19.74 m at BT15 station. The MLD at stations BT13-BT16 was all greater than 15 m and was also greater than the MLD in the Chukchi Sea shelf (Figure 9 (c)).

The M section was located to the west of the BT section, and the water depth increased from about 323 m at M11 station to 2123 m at M14 station. The MLD at M section was in the range of 5-10 m, much shallower than that at the BT section (Figure 9 (b)).



The MLD in the Chukchi Sea shelf had an overall trend of deepening from south to
north (Figure 9 (a) and (c)). The MLD in the southern Chukchi Sea shelf was smaller
than 12 m, and the MLD reached a maximum of 19.74 m in the northern Chukchi Sea
shelf. The MLD in the western continental slope was in the range of 5-10 m, similar to
that in the southern Chukchi Sea shelf. It seemed that the fluctuation of MLD in the
Chukchi Sea slope was not similar to that in the Bering Sea slope.

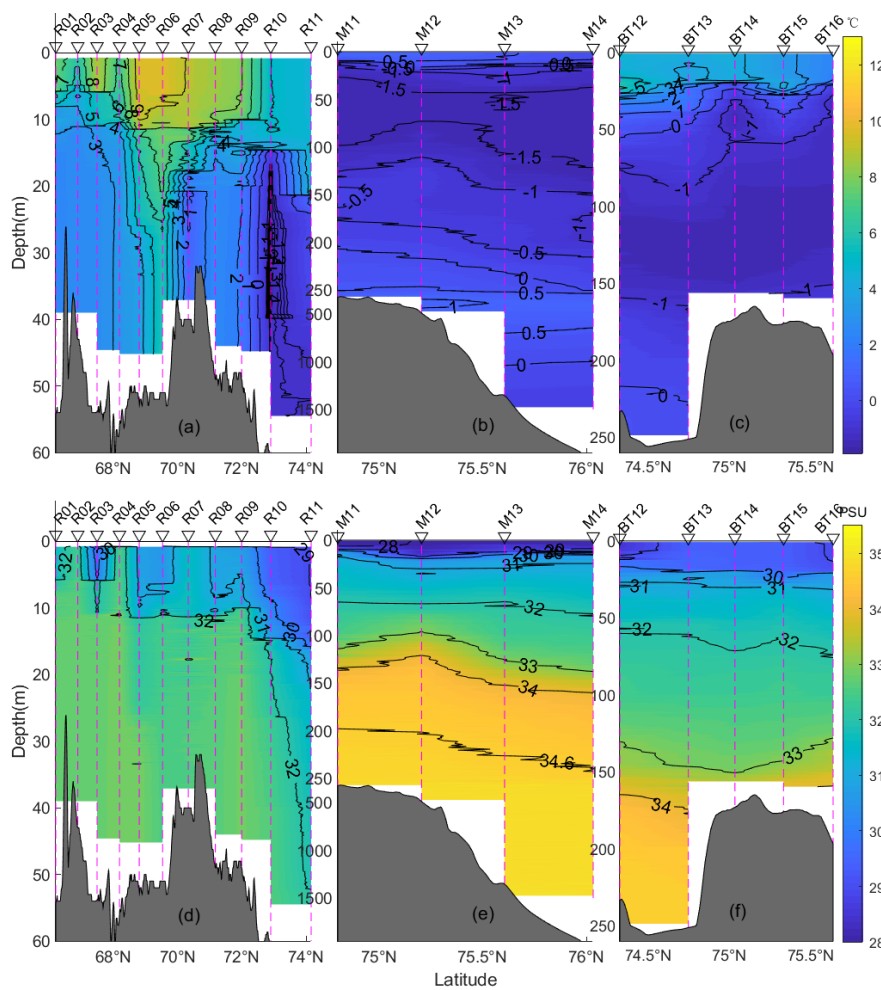


**Figure 8.** The upper panels and the lower panels show temperature and salinity
profiles, respectively. The left (a, d), middle (b, e), and right (c, f) column represent
the section of R, M, and BT, respectively.


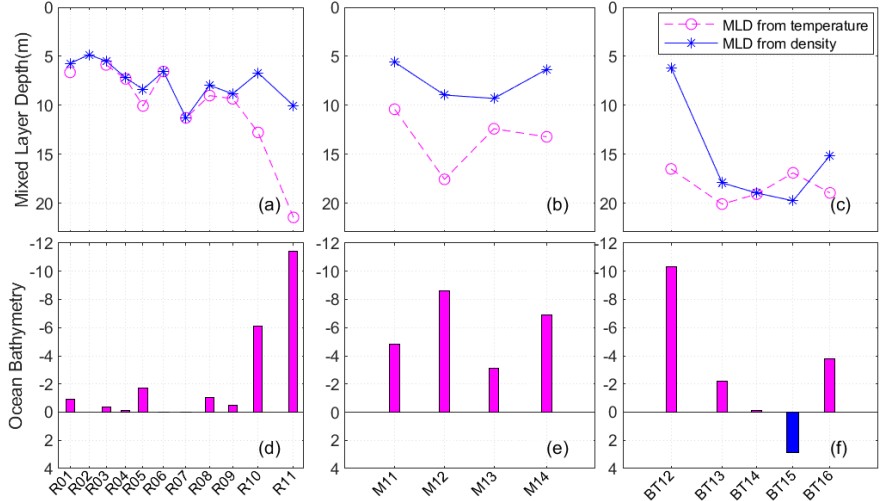

**Figure 9.** The upper panel represents the MLD from temperature and density. The lower panel represents the difference between MLDd and MLDt. The left (a, d), middle (b, e), and right (c, f) column represent the section of R, M, and BT respectively. The magenta dash lines represent the MLD calculated from the temperature, and the blue solid lines represent the MLD calculated from the density. The magenta bar means that the MLDt was larger than the MLDd, and the blue bar means that the MLDd was larger than the MLDt. Notice that the Y-axis is reversed.

In conclusion, the MLDt was different from MLDd (Figure 7 and Figure 9). The MLDd was generally larger than the MLDt in the southern Bering Sea, while the MLDt was generally larger than the MLDd in the northern Bering Sea shelf and the Chukchi Sea. The average difference between MLDd and MLDt was 0.51 m in the southern Bering Sea, and the average difference is -3.25 m in the northern Bering Sea shelf and the Chukchi Sea. The largest difference between MLDd and MLDt was less than 4 m in the southern Bering Sea, while the largest difference between MLDt and MLDd was greater than 11 m in the Chukchi Sea. The difference between MLDt and MLDd showed a tendency to increase from south to north in the Chukchi Sea.

**3.4. The relation of temperature, salinity, and MLD**

In this section, the relationships between the spatial variation of MLD and the temperature and salinity were explored.


The bottom cold water in the Bering Sea shelf was shallower than the middle cold water
mass in the Bering Sea basin. As a result, both the isopycnal and MLD were shallower
in the Bering Sea shelf than those in the Bering Sea basin. The MLD in the Bering Sea
shelf fluctuated with the topography. Therefore, the shallower MLD in the Bering Sea
shelf might be due to the terrain constraints and the bottom friction.
The isothermal and the isohaline showed a trend of deepening in the Bering Sea slope,
and the cold water mass in the middle layer also showed a trend of deepening (Figure 5
(a)). As a result, the MLD in the Bering Sea slope was larger than that in the Bering Sea
basin ( Figure 7(a)).
The density of the Anadyr Water was vertically uniform in the northwestern Bering Sea
shelf and was much larger than that of the Alaska Coastal Water in the northeastern
Bering Sea shelf. Due to the significant difference in density between the Anadyr Water
and the Alaska Coastal Water, advection occurred and the low-density water in the east
advected toward the west in the surface water, and high-density water in the west
advected toward the east at the bottom. Therefore, the seawater was stratified in the
transition zone. As a result, The MLD in the transition zone was shallower than that in
the northeastern and northwestern Bering Sea shelf (Figure 7 (c)).
The northward increase of the MLD in the Chukchi Sea was accompanied by the high
meridional gradient of the salinity and temperature. That might be the result of the
advection of the low-salinity water in the Chukchi Sea. The larger MLD at R05 and
R07 stations were related to the high-temperature and low-salinity water mass
appearing within the range of 68.5 - 70.5°N on the bottom.
Although the MLD increased in the Chukchi Sea slope as that in the Bering Sea slope,
there was a difference between them. It was remarkable that, from the ocean basin
towards the continental shelf, the isotherm and isohaline tended to parallel to the
continental slope in the Chukchi Sea, while they tended to perpendicular to the
continental slope in the Bering Sea. Therefore, it was reasonable to assume that the
reason for the deepening of the mixed layer in the Chukchi Sea slope might be different




from that in the Bering Sea slope. The changes of MLD in the Chukchi Sea slope might
be related to the low-salinity water generated from the melting of sea ice in summer and
topographical constraints. But in the Bering Sea slope, the isotherm, isohaline, and
MLD were mainly affected by the Bering Slope Current.
The MLDt was generally smaller than MLDd in the southern Bering Sea. This
indicated that the temperature constrained the mixed layer in the southern Bering Sea.
In other words, the change in density was mainly caused by the change in temperature.
The average difference between MLDd and MLDt was -3.25 m in the northern Bering
Sea and the Chukchi Sea, the absolute value of which was much greater than the 0.51 m
in the southern Bering Sea. It indicated that the MLD was dominated by the salinity in
the northern Bering Sea and the Chukchi Sea. Besides, the farther north, the greater the
difference between the MLDt and MLDd was in the Chukchi Sea, and the more
important role the salinity played in determining the MLD.
**4. Discussion on the factors influencing MLD**
**4.1. Stratification**
The contribution of the salinity and the temperature to the MLD was explored by
studying the stratification index. The stratification index covered a depth of 60 m, and
for the areas shallower than 60 m, it covered the whole water column from the sea
bottom to the surface. The temperature interpreted 30%-40% of the stratification in the
Bering Sea basin and the southern Bering Sea shelf including BL01-BL13,
BR00-BR06, as shown in Figure 10 (a) and (b). In the northeastern Bering Sea shelf
and the southern Chukchi Sea shelf, temperature interpreted 10-20% of the
stratification (Figure 10 (c) and (d)). In the northern Chukchi Sea shelf, it decreased to
5-10% (Figure 10 (f)). In the Chukchi Sea slope and the northwestern Bering Sea shelf,
the contribution of the temperature to stratification was negligible (Figure 10 (c) and
(e)). The contribution from salinity to stratification and MLD increased northward in
the upper ocean of the Bering Sea and the Chukchi Sea. This is consistent with the
previous research (Johnson et al., 2012), which showed that the seasonal variation of



the mixed layer in the Arctic was dominated by salinity. Therefore, it is reasonable to
assume that the characteristics of the mixed layer are related to the low-salinity water
generated from the melting of sea ice in the Chukchi Sea and the northern Bering Sea
shelf in the summer of 2019.

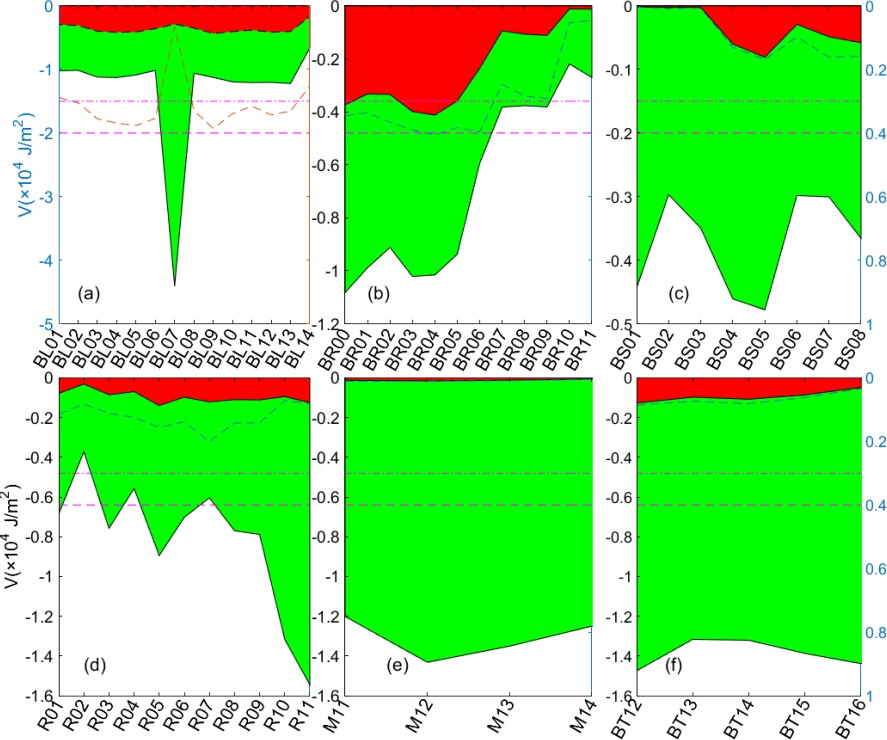


**Figure 10.** The left axis represented the stratification index. Red was the proportion of
stratification due to temperature. Green was the proportion due to salinity. The right
axis represented the proportion of the contribution of the temperature. The magenta
dash-dotted line and the dashed line represented the proportion of 30% and 40%
respectively.
**4.2. Circulation and eddy**
The deepening of the MLD in the Bering Sea slope might be related to the
strengthening of the turbulent mixing caused by the BSC and the strong vorticity along
the BSC (Figure 11 and Figure 12 (b)). As shown in Figure 11, the absolute dynamic
topography showed a high gradient along the Bering Sea slope. And the current
velocity along the Bering Sea slope was about 0.1 m/s, which was significantly larger



than that in the Bering Sea basin and the Bering Sea shelf. The large MLD at BL01 in
the northern continental slope of the Aleutian Islands was probably related to the eddies
along the Aleutian Islands (Figure 12). The MLD at BL01 was 30.04 m, significantly
larger than that at BL02, which was 18.72 m (Figure 7 (a)). The current velocity at
BL01 was about 0.2 m/s and was larger than that in the basin, which was smaller than
0.1 m/s, according to our ADCP observations.
On the basin scale, the dominant cyclonic circulation might lead to the MLD in the
central part of the Bering Sea basin smaller than that in the continental slope in the rim
of the basin.

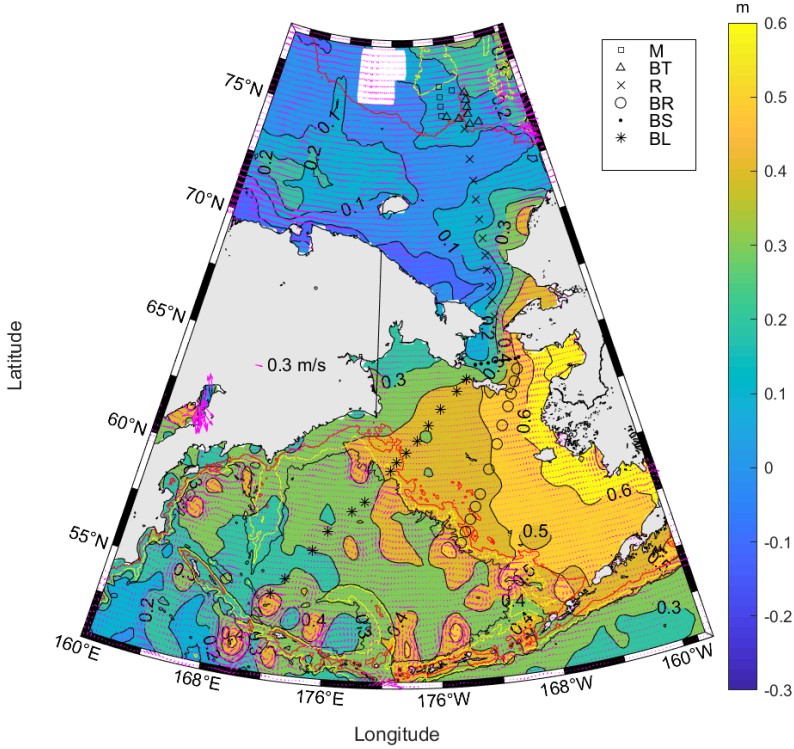


**Figure 11.** The 16-day averaged absolute dynamic topography and the surface
geostrophic flow in the Bering Sea and the Chukchi Sea from satellite altimeter.

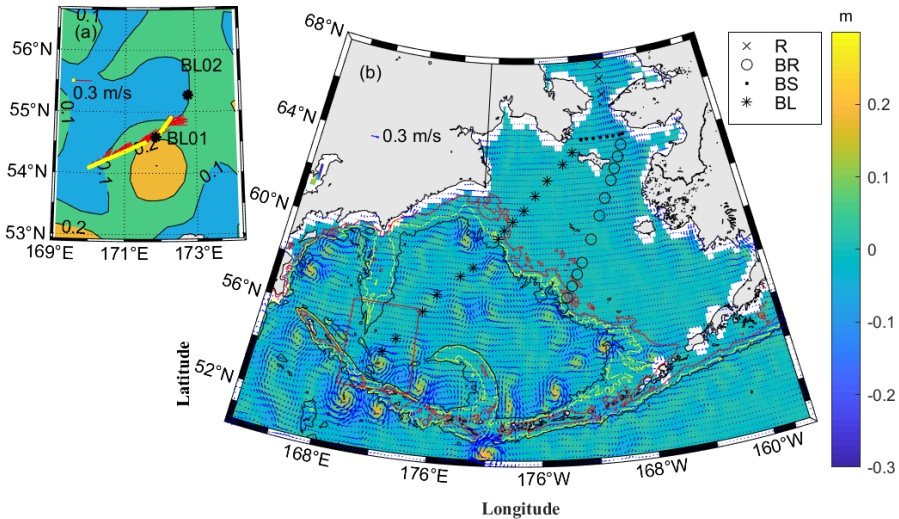

**Figure 12.** (a) The eddy next to station BL01. The contours denoted the sea surface
height from satellite observations. The yellow line denoted the track of the ship. The
red vectors denoted the ocean current velocity observed by ADCP. (b) The surface
eddy street along the Bering Sea slope from the 16-day averaged SLA. The vectors
represented the surface geostrophic flow anomaly. The color denoted the relative
vorticity normalized by the local Coriolis coefficient. The red, yellow, and blue solid
lines denoted the 200m, 2000m, and 3000m isobaths, respectively. The red rectangle
denoted the location of the region in (a).

**4.3. Momentum flux and buoyancy flux**

In summer, the Aleutian Low moved northward and the south wind prevailed. The wind
observed by the shipborne automatic meteorological station was used to assess the
CCMP wind product. The correlation coefficients of the zonal wind and meridional
wind between them were 0.92 and 0.91 respectively. And the mean difference of the
zonal wind and meridional wind between them were 0.51 m/s and 0.29 m/s respectively.
That meant the CCMP wind product behaved well in the target region.

The strengthening or weakening of stratification caused by the air-sea kinetic energy
exchange on the surface of the ocean changed the MLD. In the west of the Bering Sea
basin, in the northeast of the Bering Sea, and the north of the Chukchi Sea (BL, BS, BT,
and M stations), the MLD had a positive correlation with the wind speed, and the
correlation coefficient was 0.7 (Figure 13). The kinetic energy input into the ocean due



to the large wind speed enhanced the sea surface turbulent mixing. As a result, the MLD
increased, and BR, BT2, BL01, and BL07 were excluded. It had been known that the
MLD at BL01 and BL07 was mainly due to the influence of the continental slope
current.

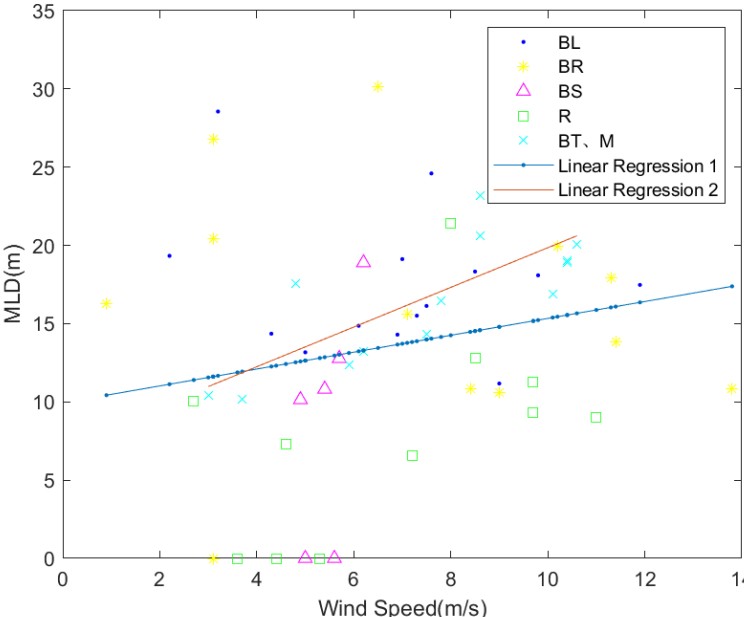


**Figure 13.** Scatter plot of wind speed and MLD. It contained BL, BS, BT, and M
stations. The solid blue line was the regression line.

The relationship between the MLD and the buoyancy flux as well as the momentum
flux was explored through Multiple Linear Regression (MLR). When the freshwater
and heat content of the upper ocean increased, the stratification was strengthened, and
the MLD decreased. When the momentum flux increased, the MLD became larger. The
average buoyancy flux caused by sea surface net heat flux and freshwater flux from
July 1 to Sept. 8 was shown in Figure 14. The momentum flux from Aug. 15 to Sept. 8
was shown as well. The correlation coefficient between the MLD and momentum flux
was 0.67, while the correlation coefficient between the MLD and buoyancy flux was
-0.33. Under the combined effect of buoyancy flux and momentum flux, the MLD
could reach a regional extremum, such as BL14, BR00, BR11, BT12, BT25, BT26,
M11, R01, R05, R08, R11 in Figure 14. The result of multiple linear regression had a
correlation coefficient of 0.41 with the measured MLD. As the yellow box in Figure 14
showed, the disappearance of the temperature mixed layer in the northwestern Bering
Sea shelf might also be related to momentum flux and buoyancy flux.

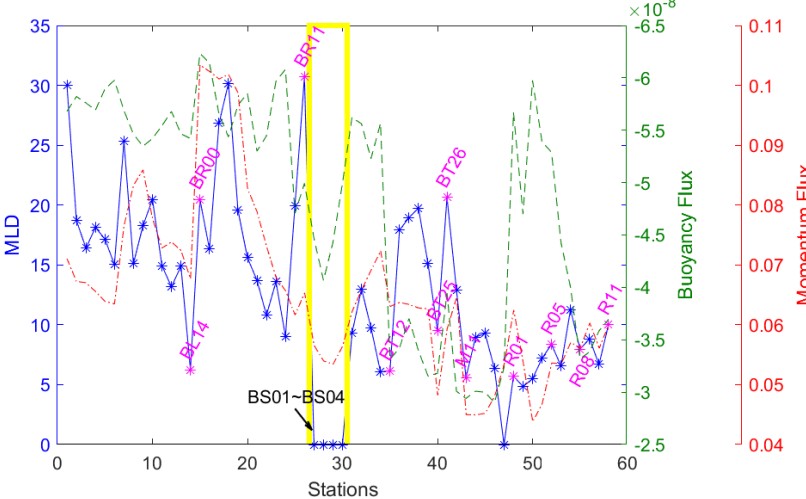


**Figure 14.** The buoyancy flux, momentum flux, and MLD of all stations. The
buoyancy flux is one-month averaged, and the momentum flux is 10-day averaged.
The order of stations is the same as Table 3.
**4.4. Waves**
It seemed that the spatial variation of MLD was not largely affected by the waves
during the expedition. It might be because the significant wave height was too small to
influence the MLD at that time, and the significant wave height was smaller than 2 m in
this region (Figure 15). The consideration of the wave had almost no improvement for
the result of the above multiple linear regression. Satellite observations showed a larger
significant wave height in the Bering Sea slope and the southeastern Bering Sea. The
significant wave height at BR00~BR04 and BL07~BL09 was larger than that in other
areas. But the MLD at BR00, BR01, BR04, BL08, BL09 was not large. The correlation
coefficient between the significant wave height and MLD was less than 0.05. The
addition of the wave made no positive contribution to the multiple linear regression
between MLD and momentum flux, buoyancy flux.



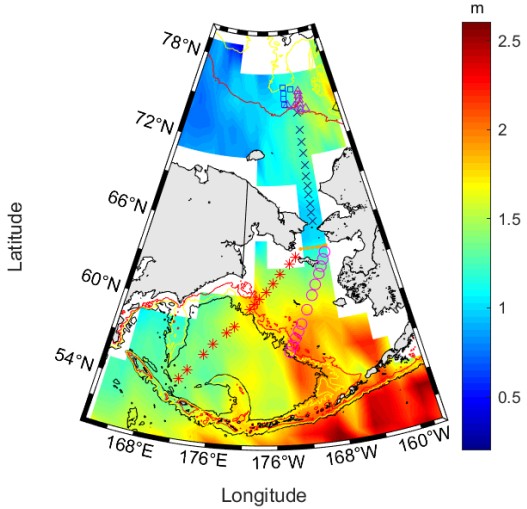

**Figure 15.** Mean significant wave height during the expedition (from Aug. 24 to Sept. 08).

**5. Conclusion**

The density threshold of $0.125 \ \mathrm{kg}/m^3$ was used to determine the MLD in the Bering Sea and the Chukchi Sea.

The in-situ data showed that the MLD in the Bering Sea basin was deeper than that in the Bering Sea shelf, but both were shallower than that in the Bering Sea slope. In the Chukchi Sea shelf, the MLD deepened from south to north. The mixed layer in the Chukchi Sea slope was similar to that in the southern Chukchi Sea shelf. The deeper mixed layer at the R05 and R07 stations was related to the high-temperature and low-salinity water masses extending from the east.

The factors that dominated the spatial variation of MLD in the Bering Sea and the Chukchi Sea were different. The MLD was constrained by the temperature and the salinity of seawater in the southern Bering Sea, while it was mainly constrained by salinity in the northern Bering Sea and the Chukchi Sea. Therefore, the upper oceanic processes relating to the MLD were different. The larger MLD in the Bering Sea slope was mainly caused by the circulation and eddies, while the MLD in the Chukchi Sea slope was mainly shaped by the spread of the northern low-salinity seawater and the



terrain constraints. The MLD in the northern Bering Sea shelf was affected by the
interaction of the high-density Anadyr Water and the low-density Alaska coastal water.
Overall, the horizontal advection led to the shallower mixed layer in the east and central
transition zone. The disappearance of the temperature mixed layer in the northwestern
Bering Sea shelf might be related to weak wind, momentum flux, and buoyancy flux.
The correlation coefficient between the momentum flux and the MLD was 0.67, larger
than that between the buoyancy flux and the MLD, which was -0.33. The correlation
coefficient between the wind and MLD reached 0.7, excluding the stations that were
obviously affected by eddy and circulation. The wave was not closely related to the
spatial variation of the MLD. The combined contributions of both the momentum flux
and the buoyancy flux interpreted the local extremum of the MLD.
**Data availability**
I would like to share my data to scientific community through Harvard Dataverse
(https://doi.org/10.7910/DVN/H07MTR). CCMP Version-2.0 vector wind analyses are
produced by Remote Sensing Systems. Data are available at www.remss.com. CFSv2
data    was    retrieved    from    NCAR    Research    Data    Archive
(https://rda.ucar.edu/datasets/ds094.0/). The sea surface height and the significant wave
height was obtained from http://marine.copernicus.eu website. The ETOPO1 was
obtained from https://www.ngdc.noaa.gov/mgg/global/global.html website.
**Author contribution**
Xiaohui Jiao performed the in situ observations, analyzed the data and prepared the
manuscript. Jicai Zhang gave expert guidance on research orientation. Chunyan Li
provided suggestions to improve the analysis and polish the language.
**Competing interests**
The authors declare that they have no conflict of interest.


**Acknowledgments**
We are extremely grateful to the chief scientist Professor Wei Zexun for his guidance
and support. We are very grateful to Liu Na, Chen Hongxia, and He Yan from the First
Institute of Oceanography for their help. This work was supported by the National Key
Research and Development Plan of China [grant number 2017YFC1404000 and
2017YFA0604100], and the National Natural Science Foundation of China [grant
number 41876086 and 41206001].





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
