# Peer review of "Observational Study on the Variability of Mixed Layer Depth in the Bering Sea and the Chukchi Sea in the Summer of 2019"

_Ocean Science, 2021_

## Referee Comment (RC1)

**Overall comments:**

This paper presents a very interesting study. The authors have excellent overall knowledge regarding the MLD, and this is clear in this work. It provides critical new data of T & S and MLD estimations of the area.

I found it very interesting to compare the MLD and MLDt and also the use of the stratification index.

As the scientific work is very good and the information that comes out of it is of great importance, I believe that the text needs to be written better to present these nice results. And also, a more robust discussion of the results is required. First, I would suggest expanding and rewriting the discussion/conclusion sessions and proceeding with professional Editing for better results.

I would also recommend using abbreviations (like the ones you have been set in Figure 1). For example, try to replace in the text the Bering Sea basin with BSb and the Bering Sea shelf with BSs or something similar (it will make it easier to for the reader)

Also, I'm not pretty sure how important it is to keep a decimal in the MLD. I don't believe that gives extra info if the MLD is 65m or 65.21m. I would round the MLDs as they are not providing any significant scientific input differences minor like that. The international references on MLDs are in meters.

Maybe Figure 11 could be left aside and just use the ADT information only in the text. Comments regarding the Figures are following.

I would suggest that this work should be published after major revisions.

**Especially for Sectors**

2.2 Data

171… and dissolved oxygen: You refer to oxygen measurements, but you are not using them anywhere in this work, so you need to make that clear or remove it from the data sector. (else the reader is waiting to see some oxygen data)

Does the data from the shipborne meteorological station were used for cross-validation of the CCMP model?  Please explain to this sector why you are presenting each dataset and how you will use it (Data and methods).

This is also valid for the fluxes and wave data etc. You are slightly presenting them in this section, but you are not making any comment regarding their use. So, we arrive at the discussion-conclusion, and we find results of correlations that have not been mentioned before. This section is the place that you'll explain your methodology:
For example: "To investigate the reasons of the spatial MLD variability, different cross-correlations and lagged (maybe) correlations have been done. First, with the wind regime ….bla bla bla. Then we have investigated the correlation of the MLD with the wave data…bla bla bla". Also, regarding the wave data, maybe it is worth it to run a Lagged Correlation to see if you obtain different results.

Also, there is an extend paragraph on this sector regarding the ADCPs, but the data and a discussion on these data is minimum in the rest of the manuscript. It looks like they belong to auxiliary data but they are really expanded in this sector. Thus, depending the use and the importance of each dataset, dedicate an appropriate paragraph in this section.

3.1.The salinity and temperature

To make it more robust, I would suggest the authors to add a small part (as you have already start doing, just extend it a bit) more dedicated to this area's water masses. There have been some studies for the Bering Sea, so a reference to these studies, regarding the water masses of the area and a comparison with the new dataset (that the authors collected in 2019) will make the manuscript more robust and complete.

3.4. The relation of temperature, salinity, and MLD

The first half of this sector, as it is written, is not providing any necessary information as it is not explaining precisely the relationship between the MLD and the T/S. Part of this info is already existing in the results. The second half (lines 447 and beyond) is written much better.

========================================================================
**More detailed comments line by line:**

What is the CCMP reanalysis? Please add reference and link

200…obtained from the CFSv2 (Sahaet al., 2011).: …what is the CFSv2, add info as you have for the Copernicus

214-228: This paragraph, as it is, is more like an Introduction part or, if better connected to the text, could be a part of the discussion regarding the paper results of estimating the MLD using different criteria, so I believe it must be or removed from the data and methods sector or rephrased in a way that will underline and explain the selected MLD estimation method of this work.

264-266: If you are referring to Kara et al. (2000), the reference depth was set at 10m, and the criterion is set as: *<the depth at the base of an isothermal (isopycnal) layer, where the temperature (density) has changed by a fixed amount of $\Delta\sigma$ $\theta$= $\sigma$ $\theta(T + \Delta T, S)$ - $\sigma$ $\theta(T, S)$, where P - 0) from the temperature (density) at a reference depth of 10 m. (with $\Delta T = 0.8°C$ )>*
You may want to refer also for the $\Delta\sigma$ $\theta = 0.125$ kg m-3 criterion to Monterey and Levitus [1997], Global Ocean (reference level 0m), Suga et al. [2004], North Pacific (ref level 10m),

☐ Monterey, G., and S. Levitus (1997), Seasonal Variability of Mixed Layer Depth for the World Ocean, NOAA Atlas NESDIS 14, 100 pp., U. S. Gov. Print. Off., Washington, D. C.
☐ Suga, T., K. Motoki, Y. Aoki, and A. M. Macdonald (2004), The North Pacific climatology of winter mixed layer and mode waters, *J. Phys. Oceanogr.*, 34, 3– 22.

…the Bering Sea basin had the characteristics of high temperature and low: … the Bering Sea basin had a high temperature and low…

300: Are you referring to the Bering Sea basin?

301: In the middle layer of the layer 50-200m? please rephrase it and give the depth that you are referring to

309-311: In the east, the density of high-temperature and low-salinity water was smaller, which had the characteristics of the Alaska Coastal Water.: High temperature and lower salinity results to lower density, so wordy writing. Do you mean that this water mass was similar to Alaska's Coastal Water? (do they have similar T, S)?.
The same also for the following lines.
Also, there is no reference to Anadyr Water. But it appears in the results without having any reference in the Introduction. Give some info for this water mass and maybe the other water masses of that area (see my previous comments regarding sector 3.1)
Please find another way to characterize the water masses that you are referring to. It's not so nice repeatedly referring to 'high-temperature and low-salinity water masses.'

354-356: Thus, the BS section represented the MLD under the influence of the advection of these two water masses: That's very interesting, so maybe you need to add some info in the Introduction section regarding the water masses in the area.

… On the contrary, the MLDt was zero there: How is that possible? I don't think it is zero. I believe that MLD and MLDt are similar because the water column looks to me (from Figure5) homogeneous. You can check that if you plot the temperature by depth. If that's so, you'll need to change it through the whole text, discussion, con conclusions, etc.

…Therefore, the shallower MLD in the Bering Sea shelf might be due to the terrain constraints and the bottom friction… please explain more or give some reference

460-461… The average difference between MLDd and MLDt was-3.25m in the northern Bering Sea and the Chukchi Sea, the absolute value of which was much greater than the 0.51 m… The absolute value what, of the MLD in the Chukchi Sea? Please refer to the station. Also, if the MLDd and MLDt difference is more or less half a meter, I'm not sure how accurate it is to tell that this demonstrated that salinity changes drive the mld. Every calculation method has an accuracy range (+- ); thus, I believe the 0.51m is in the buffer of the accuracy of the method.

491: when you are referring to the eddy, it is better to say if it cyclonic or anticyclonic

.. between them:…between the ccmp and the measured by the ship??

522-523…And the mean difference of the zonal wind and meridional wind between them were 0.51 m/s and 0.29m/s respectively…The mean difference between the meridional and zonal wind was 0.51….? is that what you mean?

532…It had been known that the MLD at BL01and BL07 was mainly due to the influence of the continental slope current... Please explain better what you mean by that and try to expand it using the appropriate references.

**Some examples of editing language issues**
============================================================
196: The sampling interval is 1 minute. : …was 1 minute (try to keep the same time through the text)

…The CCMP reanalysis wind data at the height of 10 m above the sea level was also used: ….were also used (it's plural the data), or if you preferer: the CCMP reanalysis wind dataset …was also…

…spatial resolution of CCMP data is: …of CCMP dataset.

208…bathymetric data used in this paper was from: … if you are referring to data is plural, so you use were, if you refer to a dataset you can use was

264: In what previous research are you referring to? Please specify and insert the reference.

300-304: Try to write clearer these sentences

…and the salinity was significantly lower than that in the west. In the east, the density of high-temperature and low-salinity water was smaller, which had the characteristics of the Alaska Coastal Water: …. In the east, the (high-temperature and low-salinity water) density was smaller, which had the Alaska Coastal Water characteristics.

314-316: There were…. If you are describing the data in Figure 8, then you need to be more precise; for example: at stations BL… (or at latitude…) of the Chukchi Sea shelf and the continental slope, low-density waters were present in the upper layer, with a temperature range of….
Try to write the sentence the less wordy possible.
Example for 316-322:
The temperature and salinity were gradually decreased, moving from the south to the north. At the surface, the temperature drops from 10 to 1 °C and salinity from 30 to 28. While in the bottom layers, the temperature decreased from 4 to -1.8°C and the salinity from 32 to 30….

… The temperature of the bottom water decreased from 4 to -1.3 °C from south to north, while the salinity also decreased from 32 to 30...: The bottom water temperature decreased from 4 to -1.3 °C from south to north, while the salinity also decreased from 32 to 30

321-322: There was a middle cold-water mass with a core temperature of -1.8 °Cin the depth range of 40m ~ 150m below the surface warm water in the Chukchi Sea slope.
What was the salinity of this water mass? If you can, please give a more exact position.

….was shallower than 15 m. And the minimum…: Moreover, the minimum…

…BL14 station was located in the northwestern Bering Sea: …located on the…

…which was located in the continental slope: … on the continental slope

…than all the MLD in the Bering Sea shelf: …on the Bering…

…stations in the Bering Sea shelf:…on the Bering…

352-353: The western BS section was under the influence of the water mass named Anadyr Water … The western BS section was under the influence of the Anadyr Water

… the MLD were all larger:…the MLDs were larger..

…The MLD in the continental slope of the Bering Sea was significantly:… The MLD in the Bering Sea's continental slope was significant…

… The Chukchi Sea is on the north of the Bering Strait:… is north of the Bering Strait:…

… 4.5x10-6 m

… The MLD at stations BT13-BT16 was all greater than:… BT13-BT16 was greater than

…The isothermal and the isohaline showed a trend of deepening:… The isothermal and the isohaline showed a deepening trend in the Bering Sea slope,

… than that in the Bering… than in the Bering

… advection of the low salinity water:…In what low salinity water are you referring to?

… The changes of MLD in the Chukchi Sea slope: …MLD changes in the….

… caused by the change in temperature.:…caused by temperature changes

… The contribution from salinity to:…The salinity contribution…

… research (Johnson et al., 2012): … research of Johnson et al. (2012),

500-501 … The current velocity at BL01was about 0.2m/s and was larger than that in the basin, which was smaller than 0.1m/s, according to our ADCP observations:… The current velocity at BL01was about 0.2m/s, while in the basin was measured less than 0.1m/s according to the ADCP observations.

503-505…On the basin scale, the dominant cyclonic circulation might lead to the MLD in the central part of the Bering Sea basin smaller than that in the continental slope in the rim of the basin… might lead to a smaller MLD in the central part of the Bering Sea basin, than that in the continental slope in the rim of the basin

520-521:The wind observed by the shipborne automatic meteorological station was used to assess the CCMP wind product -> this is not to be here

…and the north:…and in the north

=============================
Figures-Tables

Figure 1 or 2: It would be helpful for the reader to show in one of these figures (probably the second) the areas of the Bering Sea shelf & basin, Chukchi Sea self & slope, and the transition zone that you are referring to later in the text.
Also, please show the Anadyr Water in the Figure.

Figure 3: panel c: what are the two red boxes? Please write it to the caption.

Figure 4: Does this Figure includes all the stations? If not, it should. I suggest making one panel only, including all the stations (maybe on the vertical axis) and all the MLDts (on the horizontal axis). Also, I don't see the MLDt equal to 0 for the stations BS01-03.

Figures 5 and 9: every panel has a different depth, so please clarify the Labels for the depth (m) in each panel. Also, I would recommend adding the MLD line in every plot in these figures to make it easier for the reader to understand the MLD variability in each station.

Figure 10: it is challenging to follow. The axes' colors are mixed; in one panel the left is blue and in the next panel is black. There are the two magenta lines (explained under), but there is a red dashed line in panel a and another dashed line (probably blue) in the rest of the panels without explanation. Please make the Figure better and the captions complete.

Figure 13: …Scatter plot of the wind speed and the MLD in all the stations. …The solid blue line is the regression line of? And the red solid line? Please rewrite the caption

Figure 12: the figure and the caption are confusing, try to make them clearer.

Figure 14: explain in the caption of the Figure what is the yellow box

---

## Author Comment (AC1)

**Response to comments by Reviewers #1**

We deeply thank you for your constructive suggestions on the early version of the manuscript numbered "os-2021-7" (hereinafter named old manuscript). We have addressed all the comments formulated by the replying (in red) to your remarks (in black and blue) and the changes in manuscript (*in red italic*).

**Anonymous Referee #1**

**Overall comments:**

This paper presents a very interesting study. The authors have excellent overall knowledge regarding the MLD, and this is clear in this work. It provides critical new data of T & S and MLD estimations of the area.

I found it very interesting to compare the MLD and MLDt and also the use of the stratification index.

As the scientific work is very good and the information that comes out of it is of great importance, I believe that the text needs to be written better to present these nice results. And also, a more robust discussion of the results is required. First, I would suggest expanding and rewriting the discussion/conclusion sessions and proceeding with professional Editing for better results.

I would also recommend using abbreviations (like the ones you have been set in Figure 1). For example, try to replace in the text the Bering Sea basin with BSb and the Bering Sea shelf with BSs or something similar (it will make it easier to for the reader).

Also, I'm not pretty sure how important it is to keep a decimal in the MLD. I don't believe that gives extra info if the MLD is 65m or 65.21m. I would round the MLDs as they are not providing any significant scientific input differences minor like that. The international references on MLDs are in meters.

Maybe Figure 11 could be left aside and just use the ADT information only in the text. Comments regarding the Figures are following.

I would suggest that this work should be published after major revisions.

**Reply:**

Thanks a lot for the positive assessment and constructive comments of our paper.

The use of the stratification index was improved within the mixed layer other than within the depth of

60 m.

The result was rewrote to place the hydrographic data into the context of previous work. The discussion was expanded and rewrote by citing more references to a make it more robust. The discussion was improved to demonstrate what is new in this research.

I replaced in the text the Bering Sea basin with BSb, the Bering Sea shelf with BSs, the Bering Sea slope with BSp, Chukchi Sea shelf with CSs, Chukchi Sea slope with CSp and Bering Slope Current with BSC, as you suggested.

I round the MLDs in meters as you suggested.

*Figure 11* (numbered Figure 11 in the early version of the manuscript) was left aside and I just used the ADT information only in the text.

The 4.5 Section was supplemented in the revised manuscript to compare the temperature, salinity and MLD along the BL (Bering Sea) and R (Chukchi) sections in 2019 with previous years. The shoaling and warming of the mixed layer were found in 2019 than previous years and the climatology. And this was accompanied by the warming of the Cold Intermediate Water in the Bering Sea.

**Especially for Sectors**

**2.2 Data**

171... and dissolved oxygen: You refer to oxygen measurements, but you are not using them anywhere in this work, so you need to make that clear or remove it from the data sector. (else the reader is waiting to see some oxygen data)

**Reply:**

**Thank you for pointing this out. I am not using oxygen measurements in this work, and I have removed the description about oxygen measurements from the data sector in the revised manuscript.**

Does the data from the shipborne meteorological station were used for cross-validation of the CCMP model? Please explain to this sector why you are presenting each dataset and how you will use it (Data and methods).

This is also valid for the fluxes and wave data etc. You are slightly presenting them in this section, but you are not making any comment regarding their use. So, we arrive at the discussion-conclusion, and we find results of correlations that have not been mentioned before. This section is the place that you'll explain your methodology:

For example: "To investigate the reasons of the spatial MLD variability, different cross-correlations and lagged (maybe) correlations have been done. First, with the wind regime ....bla bla bla. Then we have investigated the correlation of the MLD with the wave data...blabla bla". Also, regarding the wave data, maybe it is worth it to run a Lagged Correlation to see if you obtain different results.

**Reply:**

Thank you for your kind suggestions. The data from the shipborne meteorological station were used to evaluate the CCMP reanalysis wind data. The wind speed bias, wind speed root-mean-square error (RMSE hereafter) of the CCMP was 1.29m, 2.37m, respectively. The temperature, salinity and pressure obtained by CTD were used to calculate MLD. The current observed from the ADCP and the sea level were used to detected eddies that might affect MLD. The wind stress and momentum flux derived from the CCMP wind data, the sea surface heat flux and water flux obtained from the CFSv2 were considered as important factors that deepen the MLD. And the detailed explanation was supplemented in the revised manuscript as following:

The wind observed by the shipborne automatic meteorological station were used to evaluate the Version 2 Cross-Calibrated Multi-Platform (CCMP) Wind Vector Analysis Product over the period from 24 Aug. to 6 Sep.. The wind speed bias, wind speed root-mean-square error (RMSE hereafter), wind direction RMSE of the CCMP wind product was 1.29m, 2.37m, and 27.46°, respectively. The correlation coefficients of the zonal wind between the CCMP wind and the measured wind by the ship were 0.92. The correlation coefficients of the meridional wind between the CCMP wind and the measured wind by the ship were 0.91. The mean difference of the zonal wind between the CCMP wind and the wind measured by the ship was 0.51 m/s. And the mean difference of the meridional wind between the CCMP wind product behaved well in the target region.

To investigate the reasons of the spatial MLD variability, different cross-correlations have been done. First, the sea level from the satellites and the ADCP observation were used to detect eddies and major ocean currents that may largely determine the MLD. The Bering Sea level was obtained by the combined measurements of several altimeters from the COPERNICUS MARINE SERVICE (The Copernicus Programme is the European Union's Earth Observation Programme, available online at https://resources.marine.copernicus.eu/?option=com csw&view=details&product id=SEALEVEL GL O PHY L4 NRT OBSERVATIONS 008 046). Second, with the wind regime, the correlation between wind speed and the MLD was explored. The relationship between the MLD and the buoyancy flux as well as the momentum flux was estimated through Multiple Linear Regression. The wind speed at the height of 10 m above the sea level and momentum flux were extracted and derived from the Version 2 CCMP (Cross-Calibrated Multi-Platform) Wind Vector Analysis Product offered by the Remote Sensing Systems (Wentz et al, 2015). The spatial resolution of CCMP dataset was 0.25°, and the temporal resolution was 6 hours. The sea surface heat flux and water flux were obtained from the National Centers for Environmental Prediction (NCEP) Climate Forecast System Version 2 (CFSv2) (Saha et al., 2011). Then we have investigated the correlation between the MLD and the wave data. The significant wave height was obtained from the COPERNICUS MARINE SERVICE (available online at https://resources.marine.copernicus.eu/?option=com csw&view=details&product id=WAVE GLO W AV L4 SWH NRT OBSERVATIONS 014 003). The bathymetric dataset used in plotting the CTD profiles was from ETOPO1 (Amante & Eakins, 2009).

Also, there is an extend paragraph on this sector regarding the ADCPs, but the data and a discussion on these data is minimum in the rest of the manuscript. It looks like they belong to auxiliary data but they are really expanded in this sector. Thus, depending the use and the importance of each dataset, dedicate an appropriate paragraph in this section.

**Reply:**

Thank you for your excellent advice. The sector regarding the ADCPs was condensed as following: CTD and lowered ADCP observations were carried out to get profiles of temperature, salinity, and velocity at these stations. The model of the CTD and ADCP were SBE 911 Plus and Teledyne RDI WHMariner 300kHz respectively (错误!未找到引用源。 and 错误!未找到引用源。 in Supporting Information). The Lamont-Doherty Earth Observatory (LDEO) software based on the inverse method (Visbeck, 2002) was used to calculate the ocean current by processing the data from the Lowered ADCP. Ship-borne ADCP measurement was carried out while the ship was in motion to get the current profile of the upper ocean along the track. The surface temperature and salinity measurements were made as well in the underway observations. The SeaBird FerryBox (**Table 1** in Supporting Information), Teledyne RDI OS 38kHz, and Teledyne RDI WHSentinel 300kHz (**Table 2** in Supporting Information) were used in the underway observations.

**3.1. The salinity and temperature**

To make it more robust, I would suggest the authors to add a small part (as you have already start doing, just extend it a bit) more dedicated to this area's water masses. There have been some studies for the Bering Sea, so a reference to these studies, regarding the water masses of the area and a comparison with the new dataset (that the authors collected in 2019) will make the manuscript more robust and complete.

**Reply:**

Thank you for your comments. A small part was added in the revised manuscript to introduce this area's water mass with more references in the revised manuscript as following:

In the northwestern Bering Sea shelf, there was a cold and salt water mass called the Anadyr Water (AW hereafter) (Wang et al., 2020; Liu et al., 2016). The Alaska Coastal Water (ACW hereafter) was located on the northeastern Bering Sea shelf with the feature of high temperature and low salinity (Wang et al., 2020; Liu et al., 2016). The layer below the surface layer was called the cold intermediate layer (CIL) in the Bering Sea basin, and it forms as a result of two processes: cooling of the water in autumn and winter and its warming in the spring and summer (Luchin et al., 1999).

The comparison between the new dataset and previous studies was supplemented in Section 4.5 in the revised manuscript as following:

**4.5 The inter-annual variation**

To explore the inter-annual variation of the MLD in the Bering Sea and Chukchi Sea, the observations along the BL section and R section from the World Ocean Atlas 2018 (WOA2018) and previous Chinese National Arctic Research Expeditions were compared.

Figure 1 The inter-annual variation of the MLD, temperature, salinity, and density of the mixed layer from the Chinese National Arctic Research Expeditions and the climatological MLD from WOA along the BL section in the Bering Sea.

---

## Author Comment (AC2)

**Response to comments by Reviewers #2**

We deeply thank you for your constructive suggestions on the early version of the manuscript numbered "os-2021-7" (hereinafter named old manuscript). We have addressed all the comments formulated by the replying (in red) to your remarks (in black) and the changes in manuscript (*in red italic*). Each picture has two numbers: the first (in red) was the order in this reply letter; the second (*in red italic*) was the order in the revised manuscript.

**Review of the manuscript:**

Observational Study on the Variability of Mixed Layer Depth in the Bering Sea and the Chukchi Sea in the Summer of 2019

Sea and the Chukem Sea in the Summer

by X Jiao, J Zhang, C Li.

I am sympathetic to oceanographers who go at sea in interesting regions of the world where climate change is amplified, such as the Bering and Chukchi Seas, and make new measurements there. There is the potential to write an interesting paper about these new measurements taken in the summer of 2019. Unfortunately, in its present state the manuscript is very far from the standards of an Ocean Science publication. Additional, more rigorous analysis and an extensive rewriting are necessary to reach the required level of quality.

**Overall comments by section**

\_\_\_\_\_

Section 1, introduction.

The introduction is not well written. It feels like a mix and match of general considerations on mixed layer dynamics, previously published results and descriptive oceanography of the region, with no clear ordering of the ideas nor focus. The "state of the art" is not presented correctly: previous studies of the mixed layer based on hydrography in your region should be mentioned in the introduction (for example, Ladd and Stabeno 2012, which you quote later in your manuscript). This section does not introduce the manuscript properly. The introduction should pose clearly each scientific question that your manuscript will attempt to answer, and explain convicingly (with recent references) why your analysis is new.

**Section 2**

The first parts, 2.1 and 2.2, are too long and wordy, and the text does not bring useful information but rather merely repeats the tables and figures. Subsection 2.3 (MLD criterion) is badly written and does

not justify clearly the choice of criterion made in the manuscript.

Section 3, results analysis. There is very little analysis in this section, the text merely describes the figures (which is unnecessary) rather than focussing on what is new, original, important. In subsection 3.1 on salinity and temperature, no reference is cited, and no attempt is made to place the hydrographic data into the context of previous work and in the context of climate change. The same for sections 3.2 and 3.3, which are too descriptive and cite no reference to previous work. The control of mixed layer depth by salinity vs. temperature is discussed in these sections, but when MLD is controlled by, say, salinity, I suppose that the stratification index is also controlled by salinity. Could you have a on temperature vs salinity control of both the MLD and the underlying stratification, to avoid repetitions? In section 3.4, the relation between temperature, salinity and MLD is discussed, but the relation with density is discussed in 4.1, this is not logical.

Section 4, factors influencing the MLD : This section is weak. It is often unclear in the text whether space variability or time variability is considered. The significance of correlations need to be computed, and the different physical mechanisms must be discussed more rigorously, based on the literature.

Section 5, Conclusion: this section is just a summary, not a conclusion. It is necessary to demonstrate what is new in your results, why they are important for the progress of Ocean Sciences, and to discuss perspectives.

**Reply:**

Thanks a lot for your assessment and constructive comments. They are valuable for improving our paper and research. I embraced your comments to present better results of our research.

Introduction was rewrote and rearranged as the following outlines: The northward heat and freshwater transport is strongly influenced by the temperature, salinity and depth of the mixed layer (Woodgate, 2018); Few works focusing on the MLD in both the Bering Sea and the Chukchi Sea were found. Most of these previous works mainly focused on the MLD at low and middle latitudes (Holte et al., 2017; Carton et al., 2008; de Boyer Montégut et al., 2004; Holte & Talley, 2009; Hosoda et al., 2010; Monterey, 1997; Schmidtko et al., 2013); Some focus on the MLD in the Arctic and found the shoaling of the MLD (Peralta-Ferriz & Woodgate (2015)); It's worth to study whether the MLD on both sides of the Bering Strait interact with each other and MLD inter-annual changes through site observation; The processes modulated the changes and distribution of the MLD in this region need to be clarified.

The Section 2.1 and 2.2 has been simplified. The lengthy description about the ADCP and CTD was deleted. The Section 2.3 was rephrased: the introduction to the different methods defining the MLD was removed the Section 1.

The result was rewrote to place the hydrographic data into the context of previous work. The discussion was expanded and rewrote by citing more references to a make it more robust. The discussion was improved to demonstrate what is new in this research.

The 4.5 Section was supplemented in the revised manuscript to compare the temperature, salinity and MLD along the BL (Bering Sea) and R (Chukchi) sections in 2019 with previous years. The shoaling and warming of the mixed layer were found in 2019 than previous years and the climatology. And this was accompanied by the warming of the Cold Intermediate Water in the Bering Sea.

**Detailled comments by line number**

(mostly on sections 1-3, I grew tired afterwards)
* * *
159: "related subjects", not "relative".

**Reply:**

Thank you for your careful inspection. I corrected the inappropriate vocabulary in the revised manuscript as "... related subjects."

164 to 171: these sentences could be clarified. How does the "air-sea kinetic energy exchange" affect the stratification? "Under the effect of wind, waves, and Langmuir circulation": wind is an atmospheric forcing, but waves and Langmuir circulations are processes taking place in the ocean, these should not be mixed up in the same sentence. Wind causes waves and Langmuir circulations but wind also causes other processes, such as vertical shear due to inertial oscillations and internal waves, that play an important part in setting the MLD. Other ocean processes such as mixed layer instabilities should be mentioned. The papers describing the results of the OSMOSIS experiment in the north earth Atlantic as especially interesting in this regard (Damerell et al 2020, and references therein).

**Reply:**

Thank you for your logical suggestion. I rephrased these sentences and corresponding references are added as well. The revised sentences were as following:

The strengthening or weakening of stratification caused by the air-sea kinetic energy exchange or buoyancy flux in the surface of the ocean will also change the MLD (Deardorff et al., 1969; Kato & Phillips, 1969; Kraus & Turner, 1967; Large et al., 1994; McWilliams et al., 1997; McWilliams et al., 2009; Price et al., 1986). Under the effect of waves, Langmuir circulation, mixed layer instabilities, and the vertical shear due to inertial oscillations and internal waves, the MLD become deeper, which has been proved by many researches based on theory, observations, and numerical models (Bruneau & Toumi, 2016; Li et al., 2013; Wu et al., 2015).

172: "In this region": which region?

**Reply:**

Thank you for pointing out my ambiguity in expression. I cleared it as following: "... in the Bering Sea and Chukchi Sea..."

178-79: "The hydrological characteristics in the Bering Sea are influenced by the Pacific Ocean due to the water exchange between the Bering Sea and the Pacific Ocean": this sentence is a bit repetitive, could the style be improved?

**Reply:**

Of course. I changed it as following: The hydrological characteristics in the Bering Sea are influenced by the Pacific Ocean due to the water

exchange, such as the major inflow through the Near Strait and outflow through Kamchatka Strait (Stabeno & Reed, 1994).

184: "Northwest wind": you mean wind from the Northwest or towards the NorthWest? Same for South wind (line 85).

**Reply:**

Northwest wind means wind from the Northwest. The same for South wind.

185: "will be frozen": the use of the future tense in this sentence is surprising.

**Reply:**

Thank you for your advice. I changed it into past tense as following: Northwest wind prevails and part of the sea surface was frozen (Zhang et al., 2010).

186-92: Explain how the subregions listed here are important for the results to be discussed in this manuscript, or else, these details are not necessary.

**Reply:**

Thank you. As these details are not necessary, I deleted these details in my revised manuscript.

189: "100m" isobath.

**Reply:**

Thank you. I corrected it and checked the mistake throughout the manuscript.

194: "The sea ice showed a trend": why the use of the past tense here? over which period is this trend observed?

**Reply:**

Markus et al. (2009) explored changes and trends in the timing of Arctic sea ice melt onset and freeze up over the period from 1979 to 2007. That's why I used the past tense here. And the manuscript was changed as following:

The sea ice in the Arctic showed a trend of later freeze up and a trend toward earlier melt onset over the period from 1979 to 2007 (Markus et al., 2009)...

196-97: The Monterey reference is too old and not specific to the region considered here. It is necessary to consider more recent references. For example, Johnson and Stabeno (2017) document the seasonal cycle of the MLD in the deep part of the Bering Sea.

**Reply:**

Thank you. I replaced the Monterey reference with the Johnson and Stabeno (2017). And the related revision in the manuscript was:

The mean MLD were around 15-20 dbar in summer and around 80-160 dbar in winter (Johnson and Stabeno, 2017).

1100, figure 1: the readability of the figure could be improved. Black text and red text are too close to each other and the red text is barely readable. In this figure as well as in the other maps of the region, readability would be much improved by using a color for continents that is outside the colorbar, such as white, grey or black.

**Reply:**

Thank you. I modified the figure 1 as you suggested: Changed the red text to make them more readable; Change the color of the continents into grey to improve the readability of the maps. The revised figure 1 was as following:

Figure 1 Figure 1. Topography, bathymetry, and circulation in the Bering Sea, Chukchi Sea, and adjacent region. Abbreviations include: ACC = Alaskan Coastal Current; SCC = Siberian Coastal Current; KC = Kamchatka Current; BSC = Bering Slope
Current; ANSC = Aleutian North Slope Current; AS = Alaskan Stream; NPC = North Pacific Current; KS=Kamchatka Strait; NS=Near Strait; AP=Amchitka Pass.
(Danielson et al., 2014; Kawaguchi & Nishioka, 2020; Johnson and Stabeno, 2017)

The color of the continents in other maps of the figure 2 in the manuscript was changed into grey as well.

---

## Author Comment (AC3)

**Response to comments by Reviewers #1**

We deeply thank you for your constructive suggestions on the early version of the manuscript numbered "os-2021-7" (hereinafter named old manuscript). We have addressed all the comments formulated by the replying (in red) to your remarks (in black and blue) and the changes in manuscript (*in red italic*). Each picture has two numbers: the first (in red) was the order in this reply letter; the second (*in red italic*) was the order in the revised manuscript.

**Anonymous Referee #1**

**Overall comments:**

This paper presents a very interesting study. The authors have excellent overall knowledge regarding the MLD, and this is clear in this work. It provides critical new data of T & S and MLD estimations of the area.

I found it very interesting to compare the MLD and MLDt and also the use of the stratification index.

As the scientific work is very good and the information that comes out of it is of great importance, I believe that the text needs to be written better to present these nice results. And also, a more robust discussion of the results is required. First, I would suggest expanding and rewriting the discussion/conclusion sessions and proceeding with professional Editing for better results.

I would also recommend using abbreviations (like the ones you have been set in Figure 1). For example, try to replace in the text the Bering Sea basin with BSb and the Bering Sea shelf with BSs or something similar (it will make it easier to for the reader).

Also, I'm not pretty sure how important it is to keep a decimal in the MLD. I don't believe that gives extra info if the MLD is 65m or 65.21m. I would round the MLDs as they are not providing any significant scientific input differences minor like that. The international references on MLDs are in meters.

Maybe Figure 11 could be left aside and just use the ADT information only in the text.
Comments regarding the Figures are following.

I would suggest that this work should be published after major revisions.

Reply:
Thanks a lot for the positive assessment and constructive comments of our paper.

The use of the stratification index was improved within the mixed layer other than within the depth of 60 m.

The result was rewrote to place the hydrographic data into the context of previous work. The discussion was expanded and rewrote by citing more references to a make it more robust. The discussion was improved to demonstrate what is new in this research.

I replaced in the text the Bering Sea basin with BSb, the Bering Sea shelf with BSs, the Bering Sea slope with BSp, Chukchi Sea shelf with CSs, Chukchi Sea slope with CSp and Bering Slope Current with BSC, as you suggested.

I round the MLDs in meters as you suggested.

*Figure 11* (numbered Figure 11 in the early version of the manuscript) was left aside and I just used the ADT information only in the text.

The 4.5 Section was supplemented in the revised manuscript to compare the temperature, salinity and MLD along the BL (Bering Sea) and R (Chukchi) sections in 2019 with previous years. The shoaling and warming of the mixed layer were found in 2019 than previous years and the climatology. And this was accompanied by the warming of the Cold Intermediate Water in the Bering Sea.

**Especially for Sectors**

2.2 Data

171… and dissolved oxygen: You refer to oxygen measurements, but you are not using them anywhere in this work, so you need to make that clear or remove it from the data sector. (else the reader is waiting to see some oxygen data)

Reply:
Thank you for pointing this out. I am not using oxygen measurements in this work, and I have removed the description about oxygen measurements from the data sector in the revised manuscript.

Does the data from the shipborne meteorological station were used for cross-validation of the CCMP model? Please explain to this sector why you are presenting each dataset and how you will use it (Data and methods).
This is also valid for the fluxes and wave data etc. You are slightly presenting them in this section, but you are not making any comment regarding their use. So, we arrive at the discussion-conclusion, and we find results of correlations that have not been mentioned before. This section is the place that you'll explain your methodology:
For example: "To investigate the reasons of the spatial MLD variability, different cross-correlations and lagged (maybe) correlations have been done. First, with the wind regime ….bla bla bla. Then we have investigated the correlation of the MLD with the wave data…blabla bla". Also, regarding the wave data,

maybe it is worth it to run a Lagged Correlation to see if you obtain different results.

Reply:

Thank you for your kind suggestions. The data from the shipborne meteorological station were used to evaluate the CCMP reanalysis wind data. The wind speed bias, wind speed root-mean-square error (RMSE hereafter) of the CCMP was 1.29m, 2.37m, respectively. The temperature, salinity and pressure obtained by CTD were used to calculate MLD. The current observed from the ADCP and the sea level were used to detected eddies that might affect MLD. The wind stress and momentum flux derived from the CCMP wind data, the sea surface heat flux and water flux obtained from the CFSv2 were considered as important factors that deepen the MLD. And the detailed explanation was supplemented in the revised manuscript as following:

*The wind observed by the shipborne automatic meteorological station were used to evaluate the Version 2 Cross-Calibrated Multi-Platform (CCMP) Wind Vector Analysis Product over the period from 24 Aug. to 6 Sep.. The wind speed bias, wind speed root-mean-square error (RMSE hereafter), wind direction RMSE of the CCMP wind product was 1.29m, 2.37m, and $27.46°$, respectively. The correlation coefficients of the zonal wind between the CCMP wind and the measured wind by the ship were 0.92. The correlation coefficients of the meridional wind between the CCMP wind and the measured wind by the ship were 0.91. The mean difference of the zonal wind between the CCMP wind and the wind measured by the ship was 0.51 m/s. And the mean difference of the meridional wind between the CCMP wind and the wind measured by the ship was 0.29 m/s. That meant the CCMP wind product behaved well in the target region.*

*To investigate the reasons of the spatial MLD variability, different cross-correlations have been done. First, the sea level from the satellites and the ADCP observation were used to detect eddies and major ocean currents that may largely determine the MLD. The Bering Sea level was obtained by the combined measurements of several altimeters from the COPERNICUS MARINE SERVICE (The Copernicus Programme is the European Union's Earth Observation Programme, available online at https://resources.marine.copernicus.eu/?option=com_csw&view=details&product_id=SEALEVEL_GLO_PHY_L4_NRT_OBSERVATIONS_008_046 ). Second, with the wind regime, the correlation between wind speed and the MLD was explored. The relationship between the MLD and the buoyancy flux as well as the momentum flux was estimated through Multiple Linear Regression. The wind speed at the height of 10 m above the sea level and momentum flux were extracted and derived from the Version 2 CCMP (Cross-Calibrated Multi-Platform) Wind Vector Analysis Product offered by the Remote Sensing Systems (Wentz et al, 2015). The spatial resolution of CCMP dataset was $0.25°$, and the temporal resolution was 6 hours. The sea surface heat flux and water flux were obtained from the National Centers for Environmental Prediction (NCEP) Climate Forecast System Version 2 (CFSv2) (Saha et al., 2011). Then we have investigated the correlation between the MLD and the wave data. The significant wave height was obtained from the COPERNICUS MARINE SERVICE (available online at https://resources.marine.copernicus.eu/?option=com_csw&view=details&product_id=WAVE_GLO_WAV_L4_SWH_NRT_OBSERVATIONS_014_003). The bathymetric dataset used in plotting the CTD profiles was from ETOPO1 (Amante & Eakins, 2009).*

Also, there is an extend paragraph on this sector regarding the ADCPs, but the data and a discussion on these data is minimum in the rest of the manuscript. It looks like they belong to auxiliary data but they are really expanded in this sector. Thus, depending the use and the importance of each dataset, dedicate

Reply:

Thank you for your excellent advice. The sector regarding the ADCPs was condensed as following:

*CTD and lowered ADCP observations were carried out to get profiles of temperature, salinity, and velocity at these stations. The model of the CTD and ADCP were SBE 911 Plus and Teledyne RDI WHMariner 300kHz respectively (**Table 1** and **Table 2** in Supporting Information). The Lamont-Doherty Earth Observatory (LDEO) software based on the inverse method (Visbeck, 2002) was used to calculate the ocean current by processing the data from the Lowered ADCP. Ship-borne ADCP measurement was carried out while the ship was in motion to get the current profile of the upper ocean along the track. The surface temperature and salinity measurements were made as well in the underway observations. The SeaBird FerryBox (**Table 1** in Supporting Information), Teledyne RDI OS 38kHz, and Teledyne RDI WHSentinel 300kHz (**Table 2** in Supporting Information) were used in the underway observations.*

3.1. The salinity and temperature

To make it more robust, I would suggest the authors to add a small part (as you have already start doing, just extend it a bit) more dedicated to this area's water masses. There have been some studies for the Bering Sea, so a reference to these studies, regarding the water masses of the area and a comparison with the new dataset (that the authors collected in 2019) will make the manuscript more robust and complete.

Reply:

Thank you for your comments. A small part was added in the revised manuscript to introduce this area's water mass with more references in the revised manuscript as following:

*In the northwestern Bering Sea shelf, there was a cold and salt water mass called the Anadyr Water (AW hereafter) (Wang et al., 2020; Liu et al., 2016). The Alaska Coastal Water (ACW hereafter) was located on the northeastern Bering Sea shelf with the feature of high temperature and low salinity (Wang et al., 2020; Liu et al., 2016). The layer below the surface layer was called the cold intermediate layer (CIL) in the Bering Sea basin, and it forms as a result of two processes: cooling of the water in autumn and winter and its warming in the spring and summer (Luchin et al., 1999).*

The comparison between the new dataset and previous studies was supplemented in Section 4.5 in the revised manuscript as following:

*4.5 The inter-annual variation*

*To explore the inter-annual variation of the MLD in the Bering Sea and Chukchi Sea, the observations along the BL section and R section from the World Ocean Atlas 2018 (WOA2018) and previous Chinese National Arctic Research Expeditions were compared.*

[Figure]

Figure 1 *Figure 15 The inter-annual variation of the MLD, temperature, salinity, and density of the mixed layer from the Chinese National Arctic Research Expeditions and the climatological MLD from WOA along the BL section in the Bering Sea.*

*The MLD in 2019 was obviously shallower and the temperature of the mixed layer was higher than those in the other five years along the BL section in the Bering Sea (**Figure 15 (a) and (b)**). This shallower MLD was accompanied by the warming of the surface layer (**Figure 6 (c) and (d)**) and cold intermediate layer (CIL) (**Figure 16 (a) ~ (f)**). The minimum temperature of the CIL water mass in the BSb showed a trend of increase: it was* 0.54℃, 0.94℃, 0.82℃, 0.69℃, 1.99℃, *and* 2.50℃ *for the year of 1999, 2003, 2010, 2012, 2014, and 2019, respectively (**Figure 16 (a)~(f)**). The warming of the CIL may be related to the air temperature warming in the previous winter and the processes in the north pacific (Overland et al., 2012).*

[Figure]

Figure 2 *Figure 16 The temperature ((a)~(f)) and salinity ((g)~(l)) profiles along the BL section in the Bering Sea from the Chinese National Arctic Research Expeditions and WOA. These expeditions were all carried out in summer.*

The MLD in 2019 was shallower than those in 1999, 2003, 2010, 2012, 2014, and 2017 along the R section in the Chukchi Sea (**Figure 17** (a)). And this was accompanied by the warming of the mixed layer (**Figure 17** (b), **Figure 8** (a) and **Figure 18**). This surface warming was related with to the regional air-sea heat flux and the Arctic amplification (Danielson et al., 2020). Chronologically, the salinity and density was

*consistent with the WOA climatological fields (**Figure 17 (c) and (d))**, while the MLD was shallower and the temperature was higher than the WOA climatological fields in the summer of 2019 (**Figure 17 (a) and (b))**. But salinity dominated the spatial fluctuation of the density for most of the year (**Figure 17 (c) and (d))**. It should be noticed that the salinity of the water in the BSs was larger than the climatology (**Figure 6 (d) and (f), Figure 15 (c))** while it was not so in the CSs (**Figure 17 (c))**. This may be linked to the increasing net glacial ablation in the Gulf of Alaska watershed (Danielson et al., 2020).*

[Figure]

Figure 3 *Figure 17 The inter-annual variation of the MLD, temperature, salinity, and density of the mixed layer from the Chinese National Arctic Research Expeditions and the climatological MLD from WOA along the R section in the Chukchi Sea.*

[Figure]

Figure 4 *Figure 18 The temperature ((a)~(f)) and salinity ((g)~(l)) profiles along the R section in the Chukchi Sea from the Chinese National Arctic Research Expeditions and WOA. These expeditions were all carried out in summer.*

**3.4. The relation of temperature, salinity, and MLD**

The first half of this sector, as it is written, is not providing any necessary information as it is not explaining precisely the relationship between the MLD and the T/S. Part of this info is already existing in the results. The second half (lines 447 and beyond) is written much better.

Thank you for your comments. The first half of this sector was rewritten to explain precisely the relationship between the MLD and the T/S and the info already existing in the results was deleted:

*In the southern BSs, the MLD at the stations BL07~BL14 and BR01~BR09 fluctuated with the topography (**Figure 5** (a) and (b)). In the north BSs, due to the significant difference in density between the Anadyr Water and the Alaska Coastal Water, advection occurred and the seawater was stratified in the transition zone. As a result, The MLD in the transition zone was shallower than that in the northeastern and northwestern BSs (**Figure 7** (c)).*

*The northward increase of the MLD in the Chukchi Sea was accompanied by the high meridional gradient of the salinity and temperature. That might be the result of the advection of the low-salinity water generated from the melting of sea ice in summer in the Chukchi Sea. The larger MLD at R05 and R07 stations might be related to the ACW appearing within the range of 68.5 - 70.5°N on the bottom.*

===========================================================================

**More detailed comments line by line:**

What is the CCMP reanalysis? Please add reference and link

Reply:
Thank you. The full name of the CCMP reanalysis data, reference and link was supplemented in the revised manuscript:

*The wind observed by the shipborne automatic meteorological station were used to evaluate the Version 2 **Cross-Calibrated Multi-Platform (CCMP) Wind Vector Analysis Product** (Wentz et al, 2015) over the period from 24 Aug. to 6 Sep..*

*The wind speed at the height of 10 m above the sea level and momentum flux were extracted and derived from the Version 2 **CCMP (Cross-Calibrated Multi-Platform) Wind Vector Analysis Product** offered by the Remote Sensing Systems (Wentz et al, 2015).*

*Version 2 **CCMP Wind Vector Analysis Product produced by Remote Sensing Systems** is available online at **http://www.remss.com/measurements/ccmp/**.*

***Wentz, F.J., J. Scott, R. Hoffman, M. Leidner, R. Atlas, J. Ardizzone, 2015: Remote Sensing Systems Cross-Calibrated Multi-Platform (CCMP) 6-hourly ocean vector wind analysis product on 0.25 deg grid, Version 2.0. Remote Sensing Systems, Santa Rosa, CA. Available online at www.remss.com/measurements/ccmp.***

200…obtained from the CFSv2 (Saha et al., 2011).: …what is the CFSv2, add info as you have for the Copernicus

Reply:

Thank you. The information for the CFSv2 was added in the revised manuscript:

*The sea surface heat flux and water flux were obtained from the National Centers for Environmental Prediction (NCEP) Climate Forecast System Version 2 (CFSv2) (Saha et al., 2011).*

*The sea surface heat flux and water flux obtained from the National Centers for Environmental Prediction (NCEP) Climate Forecast System Version 2 (CFSv2) are available online at https://rda.ucar.edu/datasets/ds094.0/.*

214-228: This paragraph, as it is, is more like an Introduction part or, if better connected to the text, could be a part of the discussion regarding the paper results of estimating the MLD using different criteria, so I believe it must be or removed from the data and methods sector or rephrased in a way that will underline and explain the selected MLD estimation method of this work.

Reply:

Thank you. This paragraph has been moved to the Introduction part.

264-266: If you are referring to Kara et al. (2000), the reference depth was set at 10m, and the criterion is set as: <the depth at the base of an isothermal (isopycnal) layer, where the temperature (density) has changed by a fixed amount of Δσ θ= σ θ(T + Δ T, S) - σ θ(T, S), where P - 0) from the temperature (density) at a reference depth of 10 m. (with ΔT = 0.8°C )>
You may want to refer also for the Δσ θ = 0.125 kg m-3 criterion to Monterey and Levitus [1997], Global Ocean (reference level 0m), Suga et al. [2004], North Pacific (ref level 10m),

☐ Monterey, G., and S. Levitus (1997), Seasonal Variability of Mixed Layer Depth for the World Ocean, NOAA Atlas NESDIS 14, 100 pp., U. S. Gov. Print. Off., Washington, D. C.
☐ Suga, T., K. Motoki, Y. Aoki, and A. M. Macdonald (2004), The North Pacific climatology of winter mixed layer and mode waters, J. Phys. Oceanogr., 34, 3– 22.

Reply:

Thank you for your advice. This paragraph has been rephrased to explain the reference depth and the suggested reference was added:

*The criterion for the MLDd was* $\Delta\sigma=0.125\text{kg}/m^3$ *, and the reference depth was 5m. The criterion was the same as some previous studies, such as Monterey and Levitus (1997) for the Global Ocean, Suga et al. (2004) for the North Pacific. But inconsistent with the reference depth of 10 m in their study, a reference depth of 5 m was adopted because the MLDd in some area was shallower than 10 m.*

296 …the Bering Sea basin had the characteristics of high temperature and low: … the Bering Sea basin had a high temperature and low…

Reply:

Thank you. I have changed the expression as following:
*The upper ocean above 30 m in the Bering Sea basin had a high temperature and low salinity pattern.*

300: Are you referring to the Bering Sea basin?

Reply:

Yes, sorry for my ambiguous sentence. I have changed it into:

*There was a cold water mass with a depth range of 50-200m and a core temperature slightly lower than 3 ℃ in the Bering Sea basin. It was called the Bering Sea Basin Intermediate Water in some studies (Liu et al., 2016).*

301: In the middle layer of the layer 50-200m? please rephrase it and give the depth that you are referring to

Reply:

I rephrased it as following in the revised manuscript:

*There was a cold water mass with a depth range of 50-200m and a core temperature slightly lower than 3 ℃ in the Bering Sea basin.*

309-311: In the east, the density of high-temperature and low-salinity water was smaller, which had the characteristics of the Alaska Coastal Water.: High temperature and lower salinity results to lower density, so wordy writing. Do you mean that this water mass was similar to Alaska's Coastal Water? (do they have similar T, S)?.

The same also for the following lines.

Also, there is no reference to Anadyr Water. But it appears in the results without having any reference in the Introduction. Give some info for this water mass and maybe the other water masses of that area (see my previous comments regarding sector 3.1)

Please find another way to characterize the water masses that you are referring to. It's not so nice repeatedly referring to 'high-temperature and low-salinity water masses.'

Reply:

Thank you for your comments. An introduction to the Anadyr Water and Alaska Coastal Water was supplemented in the revised manuscript:

*In the northwestern Bering Sea shelf, there was a cold and salt water mass called the Anadyr Water (AW hereafter) (Wang et al., 2020; Liu et al., 2016). The Alaska Coastal Water (ACW hereafter) was located on the northeastern Bering Sea shelf with the feature of high temperature and low salinity (Wang et al., 2020; Liu et al., 2016).*

And the way to characterize the water masses was changed as well in the following manuscript:

*In the northwestern Bering Sea shelf, the core temperature of the AW was about $2℃$, and the core salinity of the AW was higher than 32.5. In the northeastern Bering Sea shelf, the core temperature of the ACW was higher than $9℃$, and the salinity was significantly lower than that of AW.*

354-356: Thus, the BS section represented the MLD under the influence of the advection of these two water masses: That's very interesting, so maybe you need to add some info in the Introduction section regarding the water masses in the area.

Reply:

Thank you for your comments. Some information was added in the revised manuscript to introduce this

area's water mass with more references in the revised manuscript as following:

*In the northwestern Bering Sea shelf, there was a cold and salt water mass called the Anadyr Water (AW hereafter) (Wang et al., 2020; Liu et al., 2016). The Alaska Coastal Water (ACW hereafter) was located on the northeastern Bering Sea shelf with the feature of high temperature and low salinity (Wang et al., 2020; Liu et al., 2016). The layer below the surface layer was called the cold intermediate layer (CIL) in the Bering Sea basin, and it forms as a result of two processes: cooling of the water in autumn and winter and its warming in the spring and summer (Luchin et al., 1999).*

357 … On the contrary, the MLDt was zero there: How is that possible? I don't think it is zero. I believe that MLD and MLDt are similar because the water column looks to me (from Figure5) homogeneous. You can check that if you plot the temperature by depth. If that's so, you'll need to change it through the whole text, discussion, con conclusions, etc.

Reply:
Thank you for your comments. I agreed to your suggestions and changed it through the whole text, discussion, conclusions, etc.

428 …Therefore, the shallower MLD in the Bering Sea shelf might be due to the terrain constraints and the bottom friction… please explain more or give some reference

Reply:
Thank you for your insightful comment. I deleted this assumption in the revised manuscript. I need to do sensitivity experiment by numerical modeling to explore the evidence and the physical process in my following research.

460-461… The average difference between MLDd and MLDt was-3.25m in the northern Bering Sea and the Chukchi Sea, the absolute value of which was much greater than the 0.51 m… The absolute value what, of the MLD in the Chukchi Sea? Please refer to the station. Also, if the MLDd and MLDt difference is more or less half a meter, I'm not sure how accurate it is to tell that this demonstrated that salinity changes drive the mld. Every calculation method has an accuracy range (+- ); thus, I believe the 0.51m is in the buffer of the accuracy of the method.

Reply:
Thank you. The absolute value of the average difference between MLDd and MLDt in the southern Bering Sea. And I rephrased it in the revised manuscript:
*The average difference between MLDd and MLDt was -3.25 m in the northern Bering Sea and the Chukchi Sea, and the difference was only 0.51m in the southern Bering Sea (Including BL01~BL06, BR00).*
I removed the conclusion when the difference between the MLDd and MLDt was smaller than 1 m, in consideration of the accuracy range as you suggested.

491: when you are referring to the eddy, it is better to say if it cyclonic or anticyclonic

Reply:

Thank you. I have improved my expression to say if it cyclonic or anticyclonic throughout the revised manuscript.

523 .. between them:…between the ccmp and the measured by the ship??

Reply:

Yes. The purpose was to evaluate the CCMP wind using the measured wind by the ship. I rephrased the expression and the evaluation was supplemented with more details in the revised manuscript:

*The wind speed bias, wind speed root-mean-square error (RMSE hereafter), wind direction RMSE of the CCMP wind product was 1.29m, 2.37m, and 27.46°, respectively. The correlation coefficients of the zonal wind between the CCMP wind and the measured wind by the ship were 0.92. The correlation coefficients of the meridional wind between the CCMP wind and the measured wind by the ship were 0.91. The mean difference of the zonal wind between the CCMP wind and the wind measured by the ship was 0.51 m/s. And the mean difference of the meridional wind between the CCMP wind and the wind measured by the ship was 0.29 m/s. That meant the CCMP wind product behaved well in the target region.*

522-523…And the mean difference of the zonal wind and meridional wind between them were 0.51 m/s and 0.29m/s respectively…The mean difference between the meridional and zonal wind was 0.51….? is that what you mean?

Reply:

Sorry for my ambiguity expression. I have rephrased these sentences:

*The mean difference of the zonal wind between the CCMP wind and the wind measured by the ship was 0.51 m/s. And the mean difference of the meridional wind between the CCMP wind and the wind measured by the ship was 0.29 m/s.*

532…It had been known that the MLD at BL01and BL07 was mainly due to the influence of the continental slope current... Please explain better what you mean by that and try to expand it using the appropriate references.

Reply:

Thank you for your suggestion. The deepening the MLD at BL01 and BL07 had been discussed in Section 4.2 based on the measured current velocity and references:

*The deepening of the MLD at BL07 was related to the BSC in the BSp (Figure 11). The absolute dynamic topography showed a high gradient along the BSp. And the upper ocean current velocity at BL07 was about 0.1 m/s, which was significantly larger than that in the BSb and the BSs, according to in situ measurement. The large MLD at BL01 in the northern continental slope of the Aleutian Islands was related to the anticyclonic eddies along the Aleutian Islands (Figure 11). And this coincides with the conclusion that anticyclones deepen the MLD in the research of Gaube et al., 2019. The MLD at*

*BL01 was 30 m, significantly larger than that at BL02, which was 19 m (**Figure** 7 (a)). The upper ocean current velocity at BL01was about 0.2m/s, while it was measured less than 0.1m/s in the BSb according to the ADCP observations. The spiral of the current became irregular at the base of the mixed layer at BL01 (Figure 11 (c).*

Some examples of editing language issues
=================================================================
196: The sampling interval is 1 minute. : …was 1 minute (try to keep the same time through the text)

Reply:
Thank you. I rephrased it and checked that issue to keep the same time through the text.

197 …The CCMP reanalysis wind data at the height of 10 m above the sea level was also used: ….were also used (it's plural the data), or if you preferer: the CCMP reanalysis wind dataset …was also…

Reply:
Thank you for your patience. I corrected it as plural form: The wind speed at the height of 10 m above the sea level and momentum flux **were** extracted and derived from the Version 2 CCMP Wind Vector Analysis Product. And the whole text was checked as well.

198 …spatial resolution of CCMP data is: …of CCMP dataset.

Reply:
Thank you. Your suggestion was adopted. The revised sentence was:
The spatial resolution of CCMP dataset was 0.25°, and the temporal resolution was 6 hours.

208…bathymetric data used in this paper was from: … if you are referring to data is plural, so you use were, if you refer to a dataset you can use was

Reply:
Thank you. I corrected it into the following: The bathymetric dataset used in the CTD profiles was from ETOPO1.

264: In what previous research are you referring to? Please specify and insert the reference.

Reply:
Thank you. I have inserted the references and supplemented reasonable explanation:

*The criterion for the MLDd was* $\Delta\sigma$=0.125kg$/m^3$ *, and the reference depth was 5m.*

*The criterion was the same as some previous studies, such as Monterey and Levitus (1997) for the Global Ocean, Suga et al. (2004) for the North Pacific. But inconsistent with the reference depth of 10 m in their study, a reference depth of 5 m was adopted because the MLDd in some area was shallower than 10 m.*

300-304: Try to write clearer these sentences

Reply:
Thank you for your suggestion. I have rewritten these sentences clearer as following:
*There was a cold water mass with in the CIL and a core temperature slightly lower than 3 °C in the Bering Sea basin. The temperature of the bottom cold water mass in the southern continental shelf was similar to that of the CIL in the basin, but the bottom cold water mass was shallower due to terrain constraints on the shelf.*

As for the CIL, I have added a reference as following:
*The layer below the surface layer was called the cold intermediate layer (CIL) in the Bering Sea basin, and it forms as a result of two processes: cooling of the water in autumn and winter and its warming in the spring and summer (Luchin et al., 1999).*

308 …and the salinity was significantly lower than that in the west. In the east, the density of high-temperature and low-salinity water was smaller, which had the characteristics of the Alaska Coastal Water: …. In the east, the (high-temperature and low-salinity water) density was smaller, which had the Alaska Coastal Water characteristics.

Reply:
Thank you. I rephrased it as following:
*In the northwestern Bering Sea shelf, the core temperature of the AW was about $2℃$, and the core salinity of the AW was higher than 32.5. In the northeastern Bering Sea shelf, the core temperature of the ACW was higher than $9℃$, and the salinity was significantly lower than that of AW.*

314-316: There were…. If you are describing the data in Figure 8, then you need to be more precise; for example: at stations BL… (or at latitude…) of the Chukchi Sea shelf and the continental slope, low-density waters were present in the upper layer, with a temperature range of….

Reply:
Thank you. I rewrote them as following:
*At stations BR, M and BT of the Chukchi Sea shelf and the continental slope, low-density waters were present in the upper layer, with a temperature range of $1℃ \sim 10℃$ and salinity of $28\sim30$ **(Figure 8)**.*

Try to write the sentence the less wordy possible.

Example for 316-322:

The temperature and salinity were gradually decreased, moving from the south to the north. At the surface, the temperature drops from 10 to 1 °C and salinity from 30 to 28. While in the bottom layers, the temperature decreased from 4 to -1.8°C and the salinity from 32 to 30….

318 … The temperature of the bottom water decreased from 4 to -1.3 °C from south to north, while the salinity also decreased from 32 to 30...: The bottom water temperature decreased from 4 to -1.3 °C from south to north, while the salinity also decreased from 32 to 30

321-322: There was a middle cold-water mass with a core temperature of -1.8 °Cin the depth range of 40m ~ 150m below the surface warm water in the Chukchi Sea slope. What was the salinity of this water mass? If you can, please give a more exact position.

Reply:

Thank you for your kindness. I rewrote them as following:

*In summer, the water mass in the Chukchi Sea is a product of mixing of the Bering Sea Water and ACW (Maria N. Pisareva, 2018). On the whole, the farther north, the colder and fresher the water mass was (Pisareva, 2018). At stations BR, M and BT of the Chukchi Sea shelf (CSs hereafter) and the Chukchi Sea slope (CSp hereafter), low-density waters were present in the upper layer, with a temperature range of* $1℃～10℃$ *and salinity of* $28～30$ *(**Figure 8**). The warmer and fresher water mass around stations R05, R06, and R07 was ACW that might be diverted onto the western CSs due to the Ekman transport under the influence of anomalously strong northerly winds (Pisareva et al., 2015). And the same situation has been observed in Sep. 2004 (Pisareva, 2018), Sep. 2008 (Linders et al., 2017), Sep. 2009 (Pisareva et al., 2015), and Sep. 2012 (Pisareva, 2018). There was a cold water mass named Halocline Intermediate Water (Chen et al., 2018) with the core temperature of* $-1.8℃$ *in the depth range of 40m ~ 150m below the surface warm water in the CSp within the latitude* $74.5°N～76°N$ *(**Figure 8** (b) and (c))* .

333 ….was shallower than 15 m. And the minimum…: Moreover, the minimum…

Reply:

Thank you for your advice. I rewrote it as following:

*Moreover, the minimum of the MLD at the BL section was 6.23 m, which was observed at BL14 station.*

335 …BL14 station was located in the northwestern Bering Sea: …located on the…

Reply:

Thank you. I rewrote it as following as you suggested:

*The BL14 station was located on the northwestern Bering Sea shelf.*

338 …which was located in the continental slope: … on the continental slope

Reply:
Thank you. I rewrote it as following as you suggested:
*It occurred at the BL01 station, which was located on the continental slope on the north of the Aleutian Island.*

344 …than all the MLD in the Bering Sea shelf: …on the Bering…

Reply:
Thank you. I rewrote it as following as you suggested:
*The minimum MLD was 16.32 m at these stations and almost larger than all the MLD on the Bering Sea shelf, including the stations BR04 - BR10.*

350 …stations in the Bering Sea shelf:…on the Bering…

Reply:
Thank you. I rewrote it as following as you suggested:
*Corresponding to that, the MLD at BR10 and BR11 stations were dramatically greater than those at other BR stations on the Bering Sea shelf.*

352-353: The western BS section was under the influence of the water mass named Anadyr Water … The western BS section was under the influence of the Anadyr Water

Reply:
Thank you. I rewrote it as following as you suggested:
The western BS section was under the influence of **the Anadyr Water** in the northwestern Bering Sea, and the eastern BS section was under the influence of the Alaska Coastal Water in the northeastern Bering Sea.

355 … the MLD were all larger:…the MLDs were larger..

Reply:
Thank you. I rewrote it as following as you suggested:
*As the water column at BS01 - BS03 was well-mixed, the MLDs were larger than 35 m.*

362 …The MLD in the continental slope of the Bering Sea was significantly:… The MLD in the Bering Sea's continental slope was significant…

Reply:
Thank you. I rewrote it as following as you suggested:
*The MLD in the **Bering Sea's continental slope** was significantly greater than those in the basin and the continental shelf.*

384 … The Chukchi Sea is on the north of the Bering Strait:… is north of the Bering Strait:…

Reply:
Thank you. I rewrote it as following as you suggested:
The Chukchi Sea is north of the Bering Strait.

388 … 4.5x10-6 m

Reply:
Thank you. I rewrote it as following as you suggested:

In general, the MLD increased at a rate of $4.5 \times 10^{-6}\,\mathrm{m}$ per meter northward along

the R section (**Figure 9** (a), The rate equals $\Delta MLD(m)$ divided by distance (m). So,

the rate was dimensionless because both the units of the MLD and distance was meter.).

390 … The MLD at stations BT13-BT16 was all greater than:… BT13-BT16 was greater than

Reply:
Thank you. I rewrote it as following as you suggested:
The MLD at stations BT13-BT16 was greater than 15 m and was also greater than the MLD in the Chukchi Sea shelf (Figure 9 (c)).

430 …The isothermal and the isohaline showed a trend of deepening:… The isothermal and the isohaline showed a deepening trend in the Bering Sea slope,

Reply:
Thank you. I rewrote it as following as you suggested:
The isothermal and the isohaline showed a deepening trend in the Bering Sea slope, and the cold water mass in the middle layer also showed a deepening trend (Figure 5 (a)).

431 … than that in the Bering… than in the Bering

Reply:
Thank you. I rewrote it as following as you suggested:
As a result, the MLD in the Bering Sea slope was larger **than in the Bering Sea** basin ( Figure 7 (a)).

444 … advection of the low salinity water:…In what low salinity water are you referring to?

Reply:
Thank you. I rewrote it to make it clear:
That might be the result of the advection of the low-salinity water generated from the
melting of sea ice in summer in the Chukchi Sea.

452 … The changes of MLD in the Chukchi Sea slope: …MLD changes in the….

Reply:
Thank you.
Thank you. I rewrote it as following as you suggested:
The MLD changes in the Chukchi Sea slope might be related to the low-salinity water
generated from the melting of sea ice in summer and topographical constraints.

459 … caused by the change in temperature.:…caused by temperature changes

Reply:
Thank you. I rewrote it as following as you suggested:
In other words, the change in density was mainly caused by temperature changes.

468 … The contribution from salinity to:…The salinity contribution…

Reply:
Thank you. I rewrote it as following as you suggested:
The **salinity and the temperature contribution** to the MLD was explored by studying the
stratification index.

479 … research (Johnson et al., 2012): … research of Johnson et al. (2012),

Reply:
Thank you. I rewrote it as following as you suggested:
This was consistent with the **research of Johnson et al. (2012)**, which showed that the
seasonal variation of the mixed layer in the Arctic was dominated by salinity.

500-501 … The current velocity at BL01was about 0.2m/s and was larger than that in the
basin, which was smaller than 0.1m/s, according to our ADCP observations:… The
current velocity at BL01was about 0.2m/s, while in the basin was measured less than
0.1m/s according to the ADCP observations.

Reply:
Thank you. I rewrote it as following as you suggested:
The current velocity at BL01was about 0.2m/s, while in the basin was measured less than
0.1m/s according to the ADCP observations.

503-505…On the basin scale, the dominant cyclonic circulation might lead to the MLD in

the central part of the Bering Sea basin smaller than that in the continental slope in the rim of the basin… might lead to a smaller MLD in the central part of the Bering Sea basin, than that in the continental slope in the rim of the basin

Reply:
Thank you. I rewrote it as following as you suggested:
*On the basin scale, the dominant cyclonic circulation might lead to a smaller MLD in the central part of the Bering Sea basin, than that in the continental slope in the rim of the basin.*

520-521:The wind observed by the shipborne automatic meteorological station was used to assess the CCMP wind product -> this is not to be here

Reply:
Thank you. I moved this to section 2.2.

527 …and the north:…and in the north

Reply:
Thank you. I have added the correct preposition as you suggested:
*In the west of the BSb, the northeast of the Bering Sea, and the north of the Chukchi Sea (BL, BS, BT, and M stations), the MLD had a positive correlation with the wind speed, and the correlation coefficient was 0.6 ( the red line in Figure 12).*

==============================
Figures-Tables

Figure 1 or 2: It would be helpful for the reader to show in one of these figures (probably the second) the areas of the Bering Sea shelf & basin, Chukchi Sea self & slope, and the transition zone that you are referring to later in the text.
Also, please show the Anadyr Water in the Figure.

Reply:
Thank you. I have modified Figure 2(a) (Figure 6 in this reply letter) to show the areas of the Bering Sea shelf & basin, Chukchi Sea shelf $ slope. Figure 2(b) was modified to show the Anadyr Water, Alaska Coastal Water and the transition zone that I was referring to later in the text.

[Figure]

Figure 5 *Figure 1 Topography, bathymetry, and circulation in the Bering Sea, Chukchi Sea, and adjacent region. Abbreviations include: ACC = Alaskan Coastal Current; SCC = Siberian Coastal Current; KC = Kamchatka Current; BSC = Bering Slope Current; ANSC = Aleutian North Slope Current; AS = Alaskan Stream; NPC = North Pacific Current; KS=Kamchatka Strait; NS=Near Strait; AP=Amchitka Pass. (Danielson et al., 2014; Kawaguchi & Nishioka, 2020; Johnson and Stabeno, 2017)*

[Figure]

Figure 6 *Figure 2. (a) showed the distribution of the 58 observation stations. The asterisks, dots, circles, crosses, triangle, and squares represented the BL, BS, BR, R, BT, and M section, respectively. (b) showed the bathymetry and topography in the dashed line rectangle in (a). ACW was the abbreviation of Alaska Coastal Water.*

Figure 3: panel c: what are the two red boxes? Please write it to the caption.

Reply:
I have written the meaning of the red boxes in the caption: The local extremum in the red boxes might lead to smaller MLD than the real MLD.
And more specific quote of the red box was added in the text as well:
*BR00 was a station of type B, where the temperature of the mixed layer had local extremum, as shown by the red boxes in Figure 3(b).*

[Figure]

Figure 7 *Figure 3 Three types of temperature, salinity, and density profiles. (a), (b), and (c) showed the type A temperature, salinity, and density profiles, which had almost the same MLDt using different criteria. (d), (e), and (f) showed the type B temperature, salinity, and density profiles, and the MLDt calculated from this temperature profile using different temperature criteria was distributed around the local extremum. The local extremum in the red boxes might lead to smaller MLDt than the real MLDt. (g), (h), and (i) showed the type C temperature, salinity, and density profile; the MLDt calculated from type C temperature profile using different temperature criteria had more difference, and the distributions were more dispersed. Horizontal lines in different colors showed different MLDt responding to a group*

*of temperature criteria in (a), (d), and (g). The variable c in the legend represented the temperature criteria which ranged from 0.1 to 1 ℃. The black solid lines in (g), (h), and (i) showed the linear regression of the temperature, salinity, and density profiles within the mixed layer. The magenta (green) solid line in (i) showed density profile calculated from the depth-related temperature (salinity) and the fixed salinity (temperature) at the depth of 5 m.*

Figure 4: Does this Figure includes all the stations? If not, it should. I suggest making one panel only, including all the stations (maybe on the vertical axis) and all the MLDts (on the horizontal axis). Also, I don't see the MLDt equal to 0 for the stations BS01-03.

Reply:
Thank you for your suggestions. I modified the figure to include all the stations (on the vertical axis) and all the MLDts. The left panels for MLDt and the right panels for MLDd. The new figure 4 was shown as following:

[Figure]

Figure 8 *Figure 4. (a) The MLDt corresponding to a group of temperature criteria. The variable c in the legend represented the temperature criteria which ranged from 0.1 to 1 °C. (b) The MLDd corresponding to the criteria from the Kara et al. (2000), de Boyer Montégut et al. (2004), Holte et al (2009), and* $\Delta\sigma=0.125\mathrm{kg}/m^3$ *, respectively. Both the left and right panels were in ascending order of the latitude.*

Figures 5 and 9: every panel has a different depth, so please clarify the Labels for the depth (m) in each panel. Also, I would recommend adding the MLD line in every plot in these figures to make it easier for the reader to understand the MLD variability in each station.

Reply:
Does your "clarify the Labels for the depth(m)" mean the string "Depth(m)" near the y-axis? If so, as you have mentioned "every panel has a different depth", I changed the figure so every panel had the same depth. I used log scale y-axis. As for your "adding the MLD line in every plot", I added the MLD line as Figures bellow:

[Figure]

Figure 9 *Figure 5 The upper panels and the lower panels represented the temperature and salinity profiles, respectively. The left (a, d), middle (b, e), and right (c, f) column represented the section of*

*BL, BR, and BS, respectively. The blue solid line represented the MLDd. The magenta dashed line represented the MLDt.*

[Figure]

Figure 10 *Figure 8 The upper panels and the lower panels showed temperature and salinity profiles, respectively. The left (a, d), middle (b, e), and right (c, f) column represented the section of R, M, and BT, respectively. The blue solid line represented the MLDd. The magenta dashed line represented the MLDt.*

Figure 10: it is challenging to follow. The axes' colors are mixed; in one panel the left is blue and in the next panel is black. There are the two magenta lines (explained under), but there is a red dashed line in panel a and another dashed line (probably blue) in the rest of the panels without explanation. Please make the Figure better and the captions complete.

Reply:
Thank you for your advice. I changed the left axes' colors to black and the right axes'

colors to blue.    I uniformed the color of the lines and completed the captions as well.
The Figure 10 in the revised manuscript was in the following (Figure 11):

[Figure]

Figure 11 *Figure 10 (a)~(f) The left axis represented the stratification index. Red was the proportion of
stratification due to temperature. Green was the proportion due to salinity. The right axis represented
the percentage of the contribution of the temperature. The blue dashed line represented the proportion
of the contribution of the temperature to the stratification at different stations. (g)~(l) The mean Turner*

*Angle (Ruddick, 1983; Clement et al, 2020) within the mixed layer.*

Figure 13: …Scatter plot of the wind speed and the MLD in all the stations. …The solid blue line is the regression line of? And the red solid line? Please rewrite the caption

Reply:
Thank you. I rewrote the caption to clarify the meaning the blue solid line and the red solid line. The new caption was:
*Scatter plot of wind speed and MLD of all the stations. The red solid line was the regression line between the wind speed and the MLDD in the BL (except BL01), BR, BT, and M stations. The blue solid line was the regression line between the wind speed and the MLDD of all the stations.*
The modified Figure 13 (numbered Figure 12 in the revised manuscript) was showed in following:

[Figure]

Figure 12 *Figure 12 Scatter plot of wind speed and MLD of all the stations. The red solid line was the regression line between the wind speed and the MLDD in the BL (except BL01), BR, BT, and M stations. The blue solid line was the regression line between the wind speed and the MLDD of all the stations.*

Figure 12: the figure and the caption are confusing, try to make them clearer.

Reply:
Sorry for my fault (The title "m" in the colorbar). I corrected the figure and the caption as well. The new caption was:
*(a) The anticyclonic eddy next to station BL01 (The red rectangle in subplot (b)). The contours denoted the sea surface height from satellite observations. The yellow line denoted the track of the ship. The red vectors denoted the surface oceanic current velocity observed by ADCP. (b) The surface eddy street*

*along the Bering Sea slope from the 16-day averaged SLA. The vectors represented the surface geostrophic flow anomaly. The color denoted the vertical relative vorticity (normalized by the local planetary vorticity, f, i.e., Rossby number) at the sea surface. The red, yellow, and blue solid lines denoted the 200m, 2000m, and 3000m isobaths, respectively. The red rectangle denoted the location of the region in (a). (c) The vertical distribution of the current direction and vertical current shear at the BL01.*

The revised Figure 12 (Figure 11 in the revised manuscript as the initial Figure 11 ADT was deleted as suggested) was as following:

[Figure]

Figure 13 *Figure 12 (a) The anticyclonic eddy next to station BL01 (The red rectangle in subplot (b)). The contours denoted the sea surface height from satellite observations. The yellow line denoted the track of the ship. The red vectors denoted the surface oceanic current velocity observed by ADCP. (b) The surface eddy street along the Bering Sea slope from the 16-day averaged SLA. The vectors represented the surface geostrophic flow anomaly. The color denoted the vertical relative vorticity (normalized by the local planetary vorticity, f, i.e., Rossby number) at the sea surface. The red, yellow, and blue solid lines denoted the 200m, 2000m, and 3000m isobaths, respectively. The red rectangle denoted the location of the region in (a). (c) The vertical distribution of the current direction and vertical current shear at the BL01.*

Figure 14: explain in the caption of the Figure what is the yellow box

Reply:
As you mentioned above, it was unreasonable to say MLDt was zero. So I deleted the similar description here and the yellow box as well. The new Figure 14 (Figure 13 in the revised manuscript) in the revised manuscript was as following:

[Figure]

Figure 14 *Figure 13 The buoyancy flux, momentum flux, and MLD of all stations. The buoyancy flux was one-month averaged, and the momentum flux was 10-day averaged. The order of stations is the same was Table 3.*

---

## Author Comment (AC4)

**Response to comments by Reviewers #2**

We deeply thank you for your constructive suggestions on the early version of the manuscript numbered "os-2021-7" (hereinafter named old manuscript). We have addressed all the comments formulated by the replying (in red) to your remarks (in black) and the changes in manuscript (*in red italic*). Each picture has two numbers: the first (in red) was the order in this reply letter; the second (*in red italic*) was the order in the revised manuscript.

**Review of the manuscript:**

Observational Study on the Variability of Mixed Layer Depth in the Bering
Sea and the Chukchi Sea in the Summer of 2019
by X Jiao, J Zhang, C Li.
I am sympathetic to oceanographers who go at sea in interesting regions of the world where climate change is amplified, such as the Bering and Chukchi Seas, and make new measurements there. There is the potential to write an interesting paper about these new measurements taken in the summer of 2019. Unfortunately, in its present state the manuscript is very far from the standards of an Ocean Science publication. Additional, more rigorous analysis and an extensive rewriting are necessary to reach the required level of quality.
* * *
**Overall comments by section**
* * *
Section 1, introduction.
The introduction is not well written. It feels like a mix and match of general considerations on mixed layer dynamics, previously published results and descriptive oceanography of the region, with no clear ordering of the ideas nor focus. The "state of the art" is not presented correctly: previous studies of the mixed layer based on hydrography in your region should be mentioned in the introduction (for example, Ladd and Stabeno 2012, which you quote later in your manuscript). This section does not introduce the manuscript properly. The introduction should pose clearly each scientific question that your manuscript will attempt to answer, and explain convicingly (with recent references) why your analysis is new.

Section 2
The first parts, 2.1 and 2.2, are too long and wordy, and the text does not bring useful information but rather merely repeats the tables and figures. Subsection 2.3 (MLD criterion) is badly written and does

not justify clearly the choice of criterion made in the manuscript.

Section 3, results analysis. There is very little analysis in this section, the text merely describes the figures (which is unnecessary) rather than focussing on what is new, original, important. In subsection 3.1 on salinity and temperature, no reference is cited, and no attempt is made to place the hydrographic data into the context of previous work and in the context of climate change. The same for sections 3.2 and 3.3, which are too descriptive and cite no reference to previous work. The control of mixed layer depth by salinity vs. temperature is discussed in these sections, but when MLD is controlled by, say, salinity, I suppose that the stratification index is also controlled by salinity. Could you have a on temperature vs salinity control of both the MLD and the underlying stratification, to avoid repetitions? In section 3.4, the relation between temperature, salinity and MLD is discussed, but the relation with density is discussed in 4.1, this is not logical.

Section 4, factors influencing the MLD : This section is weak. It is often unclear in the text whether space variability or time variability is considered. The significance of correlations need to be computed, and the different physical mechanisms must be discussed more rigorously, based on the literature.

Section 5, Conclusion: this section is just a summary, not a conclusion. It is necessary to demonstrate what is new in your results, why they are important for the progress of Ocean Sciences, and to discuss perspectives.

Reply:
Thanks a lot for your assessment and constructive comments. They are valuable for improving our paper and research. I embraced your comments to present better results of our research.

Introduction was rewrote and rearranged as the following outlines: The northward heat and freshwater transport is strongly influenced by the temperature, salinity and depth of the mixed layer (Woodgate, 2018); Few works focusing on the MLD in both the Bering Sea and the Chukchi Sea were found. Most of these previous works mainly focused on the MLD at low and middle latitudes (Holte et al., 2017; Carton et al., 2008; de Boyer Montégut et al., 2004; Holte & Talley, 2009; Hosoda et al., 2010; Monterey, 1997; Schmidtko et al., 2013); Some focus on the MLD in the Arctic and found the shoaling of the MLD (Peralta-Ferriz & Woodgate (2015)); It's worth to study whether the MLD on both sides of the Bering Strait interact with each other and MLD inter-annual changes through site observation; The processes modulated the changes and distribution of the MLD in this region need to be clarified.

The Section 2.1 and 2.2 has been simplified. The lengthy description about the ADCP and CTD was deleted. The Section 2.3 was rephrased: the introduction to the different methods defining the MLD was removed the Section 1.

The result was rewrote to place the hydrographic data into the context of previous work. The discussion was expanded and rewrote by citing more references to a make it more robust. The discussion was improved to demonstrate what is new in this research.

The 4.5 Section was supplemented in the revised manuscript to compare the temperature, salinity and MLD along the BL (Bering Sea) and R (Chukchi) sections in 2019 with previous years. The shoaling and warming of the mixed layer were found in 2019 than previous years and the climatology. And this was accompanied by the warming of the Cold Intermediate Water in the Bering Sea.
* * *
**Detailled comments by line number**

(mostly on sections 1-3, I grew tired afterwards)
* * *
l59: "related subjects", not "relative".

Reply:
Thank you for your careful inspection. I corrected the inappropriate vocabulary in the revised manuscript as *"… related subjects."*

l64 to l71: these sentences could be clarified. How does the "air-sea kinetic energy exchange" affect the stratification? "Under the effect of wind, waves, and Langmuir circulation": wind is an atmospheric forcing, but waves and Langmuir circulations are processes taking place in the ocean, these should not be mixed up in the same sentence. Wind causes waves and Langmuir circulations but wind also causes other processes, such as vertical shear due to inertial oscillations and internal waves, that play an important part in setting the MLD. Other ocean processes such as mixed layer instabilities should be mentioned. The papers describing the results of the OSMOSIS experiment in the north earth Atlantic as especially interesting in this regard (Damerell et al 2020, and references therein).

Reply:
Thank you for your logical suggestion. I rephrased these sentences and corresponding references are added as well. The revised sentences were as following:
*The strengthening or weakening of stratification caused by the air-sea kinetic energy exchange or buoyancy flux in the surface of the ocean will also change the MLD (Deardorff et al., 1969; Kato & Phillips, 1969; Kraus & Turner, 1967; Large et al., 1994; McWilliams et al., 1997; McWilliams et al., 2009; Price et al., 1986). Under the effect of waves, Langmuir circulation, mixed layer instabilities, and the vertical shear due to inertial oscillations and internal waves, the MLD become deeper, which has been proved by many researches based on theory, observations, and numerical models (Bruneau & Toumi, 2016; Li et al., 2013; Wu et al., 2015).*

l72: "In this region": which region?

Reply:
Thank you for pointing out my ambiguity in expression. I cleared it as following:
*"… in the Bering Sea and Chukchi Sea…"*

l78-79: "The hydrological characteristics in the Bering Sea are influenced by the Pacific Ocean due to the water exchange between the Bering Sea and the Pacific Ocean": this sentence is a bit repetitive, could the style be improved?

Reply:
Of course. I changed it as following:
*The hydrological characteristics in the Bering Sea are influenced by the Pacific Ocean due to the water exchange, such as the major inflow through the Near Strait and outflow through Kamchatka Strait (Stabeno & Reed, 1994).*

l84: "Northwest wind": you mean wind from the Northwest or towards the NorthWest? Same for South wind (line 85).

Reply:
Northwest wind means wind from the Northwest. The same for South wind.

l85: "will be frozen": the use of the future tense in this sentence is surprising.

Reply:
Thank you for your advice. I changed it into past tense as following:
*Northwest wind prevails and part of the sea surface was frozen (Zhang et al., 2010).*

l86-92: Explain how the subregions listed here are important for the results to be discussed in this manuscript, or else, these details are not necessary.

Reply:
Thank you. As these details are not necessary, I deleted these details in my revised manuscript.

l89: "100m" isobath.

Reply:
Thank you. I corrected it and checked the mistake throughout the manuscript.

l94: "The sea ice showed a trend": why the use of the past tense here? over which period is this trend observed?

Reply:
Markus et al. (2009) explored changes and trends in the timing of Arctic sea ice melt onset and freeze up over the period from 1979 to 2007. That's why I used the past tense here. And the manuscript was changed as following:
*The sea ice in the Arctic showed a trend of later freeze up and a trend toward earlier melt onset over the period from 1979 to 2007 (Markus et al., 2009)…*

l96-97: The Monterey reference is too old and not specific to the region considered here. It is necessary to consider more recent references. For example, Johnson and Stabeno (2017) document the seasonal cycle of the MLD in the deep part of the Bering Sea.

Reply:
Thank you. I replaced the Monterey reference with the Johnson and Stabeno (2017). And the related revision in the manuscript was:
The mean MLD were around 15-20 dbar in summer and around 80-160 dbar in winter (Johnson and Stabeno, 2017).

l100, figure 1: the readability of the figure could be improved. Black text and red text are too close to each other and the red text is barely readable. In this figure as well as in the other maps of the region, readability would be much improved by using a color for continents that is outside the colorbar, such as white, grey or black.

Reply:
Thank you. I modified the figure 1 as you suggested: Changed the red text to make them more readable; Change the color of the continents into grey to improve the readability of the maps. The revised figure 1 was as following:

[Figure]

Figure 15 *Figure 1. Topography, bathymetry, and circulation in the Bering Sea, Chukchi Sea, and adjacent region. Abbreviations include: ACC = Alaskan Coastal Current; SCC = Siberian Coastal Current; KC = Kamchatka Current; BSC = Bering Slope Current; ANSC = Aleutian North Slope Current; AS = Alaskan Stream; NPC = North Pacific Current; KS=Kamchatka Strait; NS=Near Strait; AP=Amchitka Pass. (Danielson et al., 2014; Kawaguchi & Nishioka, 2020; Johnson and Stabeno, 2017)*

The color of the continents in other maps of the figure 2 in the manuscript was changed into grey as well.

[Figure]

Figure 16 *Figure 2. (a) showed the distribution of the 58 observation stations. The asterisks, dots, circles, crosses, triangle, and squares represented the BL, BS, BR, R, BT, and M*

*transection, respectively. (b) showed the bathymetry and topography in the dashed line rectangle in (a). ACW was the abbreviation of Alaska Coastal Water.*

l107-108: the Monterey dataset is older than ARGO. Please also mention the Holte dataset in this list.

Reply:
Thank you for your recommendation. I mentioned the Holte dataset in the revised manuscript as following:
*Thanks to the rapid growth of Argo observations in the past decade, the MLD in most of the global ocean has been better studied (**Holte et al., 2017**). There are several global MLD datasets available (Carton et al., 2008; de Boyer Montégut et al., 2004; **Holte & Talley, 2009**; Hosoda et al., 2010; Monterey, 1997; Schmidtko et al., 2013).*

l120-123: please avoid casual style. The enumeration "will benefit the model calibration and evaluation, air-sea interaction, and climate change, etc." is not fit for a scientific paper, unless you establish precisely how your paper will impact each of these different scientific domains.

Reply:
Thank you for your criticism. I deleted that enumeration in the revised manuscript and I will pay attention to this point in my future research paper as well. The corrected line 120-123 was as following:
*In this paper, the field observational data sampled during the summer of 2019 will be analyzed to study the spatial variations of MLD in the Bering Sea and the Chukchi Sea.*

l128-140: "2.1 study area" presents only the bathymetry. Why is it important to list the depths of all the subregions in the text? A look at the maps of figure 2 is enough (although figure 2 could be improved). This subsection 2.1 seems unnecessary.

Thank you for pointing out my unnecessary subsection. In consideration of the introduction to the circulation, wind, sea-ice, etc. in this region, I agree with your suggestion. I deleted the subsection 2.1 in my revised manuscript.

l144: what is the meaning of the section designations (BL, BR, BS, R, BT, and M)? Do the letters refere to something?

Reply:
The letters are meaningless and refer to nothing. And a supplementary note was made in the revised manuscript:
*As shown in Figure 2, 58 stations were distributed in BL, BR, BS, R, BT, and M section (The section designations are meaningless and refer to nothing.).*

l145-150: it is not necessary to repeat the location of the sections in the text. The figure is enough.

Reply:

Thank you. The repetitive part of the location of the sections in the text was deleted in the revised manuscript.

l149-150 "These sections are representatives of this region": what do you mean by "representative"? representative of different bathymetries? different hydrography? current regimes? Certainly they are not representative of the seasonal cycle, being taken in summer only.

Reply:

Thank you for pointing it out. I changed it into a more rigorous style as following:

*These sections are representatives of different bathymetries, hydrography, and current regimes during the expedition period in this region.*

l151-183: This subsection 2.2 is redundant with the tables. If you keep the tables, you can shorten this text and avoid listing technical details such as the reference of the equipment, sampling details, etc which the reader can find in the tables. You can replace this text by a short paragraph pointing to what is new and original. For example, have hydrographic measurements been carried out in this region before? Are such measurements available in distributed databases such as World Ocean Atlas (WOA), of EN4? In which way do your measurements complement these existing databases? Are there ADCP data already available in this regions? In which way is your dataset new and different?

Reply:

Thank you. I kept the tables and deleted the repetitive text about technical details which can be found in the tables. And the following was supplemented to the revised manuscript to explain in which way this dataset was new and different:

*The dataset was valuable as there were no such measurements for the summer of 2019 available in distributed databases such as Word Ocean Atlas (WOA) when this manuscript was submitted. As similar ADCP and CTD measurements were performed along the BL, BS, and R during the previous Chinese National Arctic Research Expedition, the inter-annual variation of the hydrography and MLD could be explored.*

l184-196: the two tables 1 and 2 about the details of the equipment could be merged into one table.

Reply:

Thank you. The CTD and ADCP were listed for different technical details, so the table were different and difficult to merged into one table. If they were burdensome in the manuscript, I will move them into the supplemental information.

l187, table 3: it is not usual to list longitude and time of each hydrographic station in XXIst century oceanographic papers. This information is usually shown on a map (which you do in figure 2) and the actual numbers are found in the databases or in the supporting datasets made available with the manuscript. Table 3 is not necessary.

Reply:

Thank you. I moved them into the supporting datasets available with the manuscript.

l199, figure 2: the figure could be more readable (see remark about figure 1). If you want to point out some isobaths, please superimpose the corresponding contours, or use a discrete colorbar.

Reply:
Thank you. The color of the continents in other maps of the figure 2 in the manuscript was changed into grey as well.

[Figure]

Figure 17 *Figure 2. (a) showed the distribution of the 58 observation stations. The asterisks, dots, circles, crosses, triangle, and squares represented the BL, BS, BR, R, BT, and M transection, respectively. (b) showed the bathymetry and topography in the dashed line rectangle in (a). ACW was the abbreviation of Alaska Coastal Water.*

l197: please spell out what CCMP means.

Reply:

Thank you. I spelt out CCMP, CFSv2, etc. The corresponding revised sentence was as following:

*The wind observed by the shipborne automatic meteorological station were used to evaluate the Version 2 Cross-Calibrated Multi-Platform (CCMP) Wind Vector Analysis Product.*

l200: please spell out what CFSv2 means.

Reply:

Thank you. I spelt out CFSv2. The first sentence in which CFSv2 appeared was supplemented as following:

*The sea surface heat flux and water flux were obtained from the National Centers for Environmental Prediction (NCEP) Climate Forecast System Version 2 (CFSv2) (Saha et al., 2011, available online at https://rda.ucar.edu/datasets/ds094.0/).*

l 203-207: please quote the publications describing these copernicus datasets. The links to the web sites should appear in the "data availability" section, not in the text.

Reply:

Thank you. I noticed that issue while writing the manuscript. So I checked the terms ( as shown in Figure 18) in the "How to cite or reference Copernicus Marine Products and Services?":

It will depend on the item but **in general**, you will find the information on **how to cite and reference** a **product/dataset** on its Copernicus Marine **Catalogue Entry** (see **"References" block** under **"Information" Tab**, when available):

[Figure]

Figure 18 The example of indicating where to find the references in the terms.

The truth was that some of the dataset products have no reference in the dataset information catalogue entry and so does the dataset I used (as shown in Figure 19):

[Figure]

Figure 19 The information tab of the dataset I used in the paper.

l214-216: why do you quote examples from two old papers (Smyth et al, 1996 and Wijesekear et al, 1996) rather than give more details on the methods used in more recent papers such as de Boyer Montegut, Holte, etc?

Reply:
Thank you.
For the comments on l214-216, l218-219, l219-221, l221-l224, l224-227, I deleted this part while replacing the old papers with the more recent papers.
For the comments on l214-216 and l218-219, I replaced the old papers with the more recent papers in the revised manuscript, and moved this paragraph to the Section 1 Introduction:
*Methods to estimate MLD include difference threshold (de Boyer Montégut et al., 2004; Kara et al., 2000; Kara et al., 2003), gradient threshold (Lukas & Lindstrom, 1991), curvature method (Lorbacher et al., 2006), split and merge method (Thomson & Fine, 2003), hybrid method (Holte et al., 2009), etc. For example, Kara et al. (2000, 2003) defined the Isothermal Layer Depths (ILD) as being the depth at the base of an isothermal layer, where the temperature has changed by a fixed amount of $\Delta T$ from the temperature at a reference depth of 10 m, and the mixed layer depth (MLD) was the depth at the base of an isopycnal layer where the density has changed by a fixed amount of*

$$\Delta \sigma_t = \sigma_t(T + \Delta T, S, P) - \sigma_t(T, S, P)$$ *from the density at a reference depth of 10 m. Note that their*

*$\Delta\sigma$ criterion varied based on a fixed $\Delta T$. de Boyer Montégut et al. (2004) defined the MLD as the depth within which the temperature (density) varied within a threshold value of $\Delta T = 0.2°C$ ($\Delta\sigma = 0.03 \text{kg}/m^3$) relative to the value at 10 m depth. Some researchers proposed a split-and-merge method, which could be used not only to calculate the MLD but also to describe other marine vertical structural features (Thomson & Fine, 2003). Holte et al (2009) came up with a hybrid method, which derived five possible MLD values for density profiles: the density threshold MLD estimate, the density gradient MLD estimate, and the intersection of the density mixed layer and thermocline fits, as well as the temperature threshold MLD estimate, collocated temperature and temperature gradient maxima, the temperature maximum, and the final MLDs from the temperature and salinity algorithms, and then analyzed the patterns in the suite to select a final MLD estimate.*

l218-219 "many researchers used a gradient threshold of 0.1 kg/ð'š4 (Lukas & Lindstrom, 1991)". Why this old reference? Please discuss the most recent methods, starting with Kara (2000, 2003), Clement de Boyer Montegut (2004) or Holte et al (2009).

*Reply:*
*Thank you. As this comment on l218-219 focused on the old papers as the comment above (on l214-l216), and were replied after the comments on l214-216.*

l219-221 :What is the "least-squares regression and integration method" and who invented it or used it? Is this relevant for your manuscript?

*Reply:*
*Thank you. I deleted these methods from old reference, and discussed the most recent methods as mentioned in the reply to the comment on l214-216.*

l221-l224 : "Some researchers proposed a split-and-merge method, which could be used not only to calculate the MLD but also to describe other marine vertical structural features (Thomson & Fine, 2003). Therefore, the difference threshold and gradient threshold are better choices.". When you use "Therefore" to start a sentence, it means that your statement is a consequence of the previous sentences. Here, the preceeding sentences do not demonstrate in any way why the difference treshold and gradient are better.

*Reply:*
*Thank you. I deleted these methods from old reference, and discussed the most recent methods as mentioned in the reply to the comment on l214-216.*

l224-227: provide a reference where it is demonstrated that dissolved oxygen is not an accurate method.

*Reply:*
*Thank you. I deleted these methods from old reference, and discussed the most recent methods as mentioned in the reply to the comment on l214-216.*

l235-237: "the temperature of the mixed layer had local extremum. As a result, if a small threshold was

used, the calculated MLD would be shallower than the real MLD." What is the "real" MLD? By definition, the MLD is the depth over which everything can be considered "well-mixed" (temperature, density, salinity). If temperature is not mixed, then you have not defined a "true" or "real" MLD. Please show the corresponding profiles of salinity and density to demonstrate that they are indeed mixed.

Reply:

Thank you. The local extremum means that due to the high accuracy of the instruments, the temperature, salinity, and density profiles are not pretty smooth and may have very small fluctuation within the MLD. I have added the profiles of salinity and density in the revised manuscripts as following:

[Figure]

Figure 20 **Figure 3.** *Three types of temperature, salinity, and density profiles. (a), (b), and (c) showed the type A temperature, salinity, and density profiles, which had almost the same MLDt using different criteria. (d), (e), and (f) showed the type B temperature, salinity, and density profiles, and the MLDt calculated from this temperature profile using different temperature criteria was distributed around the local extremum. The local extremum in the red boxes might lead to smaller MLDt than the real MLDt. (g), (h), and (i) showed the type C temperature, salinity, and density profile; the MLDt calculated from type C temperature profile using different temperature criteria had*

*more difference, and the distributions were more dispersed. Horizontal lines in different colors showed different MLDt responding to a group of temperature criteria in (a), (d), and (g). The variable c in the legend represented the temperature criteria which ranged from 0.1 to 1 °C. The black solid lines in (g), (h), and (i) showed the linear regression of the temperature, salinity, and density profiles within the mixed layer. The magenta (green) solid line in (i) showed density profile calculated from the depth-related temperature (salinity) and the fixed salinity (temperature) at the depth of 5 m. The upward-pointing triangle, downward-pointing triangle, square, and asterisk in (f) showed the MLDd got based on the criteria of Kara et al. (2000), de Boyer Montégut et al. (2004), Holte et al (2009), and* $\Delta\sigma = 0.125\text{kg} / m^3$.

l230-255: It is unclear what your types A, B, C are. Please explain at the beginning of this section how you classify the profiles, providing equations if necessary. The way the text is written, at the beginning your classification of profiles into categories seems to be based only on temperature (Figure 3) while in fact you end up choosing a density-based threshold and you show that salinity is important. All this discussion has to be rethought carefully and rewritten completely. Please classify the profiles as a function of their control by salinity or temperature, and show the profiles of density, salinity and temperature in figure 3.

Reply:
I rewrote this paragraph to clarify what types A, B, C are as following. The figure was modified (as shown in Figure 20) to include the temperature, salinity, and density profiles in the revised manuscript.

*According to the shapes of the temperature, salinity, and density profiles, they were identified into three classes: type A profiles within the mixed layer were almost completely homogenous, and showed no gradient and fluctuation; type B profiles showed obvious fluctuation, as shown in the red box in (d), (e), and (f) of Figure 3; type C profiles showed both obvious gradient (black line in (g), (h), and (i) of Figure 3) and fluctuations within the mixed layer. BR 01, BR00, and BL08 showed the profiles of the temperature, salinity, and density of type A, B, C, respectively (Figure 3). Due to the existence of the fluctuations (in the red box in (d), (e), and (f) of Figure3) in the temperature, salinity, and density profiles, suitable criteria were required to get MLD.*

l258-259: you mention a criterion (0.5) for temperature but not for density, the sentence is illogical.

Reply:

Thank you. I mentioned a criterion ( $\Delta\sigma = 0.125\text{kg} / m^3$ ) for density later in the initial manuscript:

*The MLDd was defined as the depth at which **density** differed from that of the depth of 5 m by 0.125* $\text{kg} / m^3$.

To make it clear, I rewrote these relevant sentences in the revised manuscript:

*The criterion for the **MLDd** was* $\Delta\sigma = 0.125\text{kg} / m^3$, *and the reference depth was 5m. The criterion*

*was the same as some previous studies, such as Suga et al. (2004) for the North Pacific. But inconsistent with the reference depth of 10 m in their study (Suga et al., 2004; de Boyer Montégut et al., 2004), a reference depth of 5 m was adopted because the MLDd in some area was shallower than 10 m. The suitable criterion for the **temperature difference method** was 0.5 $°C$ , and the reference depth was also 5m.*

l263-265: Here for the first time you explain what "threshold" means and you say that you look at the difference between density at a given depth and density at 5m. Do you also consider temperature at 5m? Why 5m, while others such as De Boyer Montegut use 10m? This information should come earlier.

Reply:

Thank you for your suggestion. The Section 2.2 was modified significantly. The threshold method was explained at the beginning of Section 2.2:

*In this paper, the most widely adopted **difference threshold method** (de Boyer Montégut et al., 2004) was used to estimate the MLD. The MLDd was defined as the depth at which density differed from that of the reference depth by a criterion.*

I did also consider temperature at 5 m, because the MLDd and MLDt in some area was shallower than 10 m. And this was explained as well in the revised manuscript:

*The criterion for the MLDd was $\Delta\sigma=0.125\text{kg}/m^3$ , and the reference depth was 5m. The criterion was the same as some previous studies, such as Suga et al. (2004) for the North Pacific. But inconsistent with the reference depth of 10 m in their study (Suga et al., 2004; de Boyer Montégut et al., 2004), **a reference depth of 5 m was adopted because the MLDd in some area was shallower than 10 m**. The suitable criterion for the **temperature difference method was 0.5 $°C$ , and the reference depth was also 5 m**.*

l264: "This is consistent with previous research": which research? Certainly not Clement de Boyer or Holte who use lower density jumps. Can you justify your choice by comparing the different methods using your data, rather than relying arbitrarily on one publication, Kara 2000?

Thank you for your suggestion. I supplemented the reference in the revised manuscript:

*The criterion for the MLDd was $\Delta\sigma=0.125\text{kg}/m^3$ , and the reference depth was 5m. The criterion was the same as some previous studies, such as **Suga et al. (2004)** for the North Pacific.*

And the MLDd got based on the criteria of Kara et al. (2000), de Boyer Montégut et al. (2004), Holte et al (2009), and $\Delta\sigma=0.125\text{kg}/m^3$ . were marked and compared in (f) of Figure 20 (which was Figure 3 in the revised manuscript).

l242: figure 3. Please show temperature, density and salinity profiles. Please indicate the location of the profiles you have chosen.

Reply:

Thank you for your suggestion. I supplemented temperature, salinity, and density profiles in Figure 3 in the revised manuscript (Also shown in Figure 20 of this reply letter). The location of the profiles I have chosen were indicated in Figure 16 of this reply letter (Figure 2 in the revised manuscript).

l268, figure 4: Please have a horizontal axis in kilometers besides the stations labels, or else, because your sections are mainly oriented south/north, use the latitude. What are the different criteria listed?

Reply:
Thank you for your suggestions. I modified the figure in the revised manuscript to have an axis in latitude (on the vertical axis). The left panels showed the MLD from temperature and the right panels showed the MLD from density. The new figure 4 was shown as following:

[Figure]

Figure 21 **Figure 4.** *(a) The MLDt corresponding to a group of temperature criteria. The variable c in the legend represented the temperature criteria which ranged from 0.1 to 1 ℃. (b) The MLDd corresponding to the criteria from the Kara et al. (2000), de Boyer Montégut et al. (2004), Holte et al (2009), and $\Delta\sigma$=0.125kg / $m^3$, respectively. Both the left and right panels were in ascending order of the latitude.*

l 274-288: The stratification criterion is not relevant for the mixed layer if you compute it over the entire depth of the water column where the ocean is deep. Ladd and Stabeno compute it down to 60m. Please explain here what you do exactly, and why you choose 60m. You may also write an equation to show how you compute the relative contributions of temperature and salinity. You won't have to repeat the method in section 4.

Reply:

Thank you for your suggestion. I modified the manuscript and the stratification index was calculated within the MLD. The figure 10 was revised and Section 4.1 was rewrote as well, as shown in the following. The contribution to SI due to temperature (SIt) and Salinity (SIs) was SIs/(SIt+SIs), SIt/(SIt+SIs), respectively.

*4.1. Stratification*

*The salinity and the temperature contribution to the MLD was explored by studying the stratification index (SI). The SI covered the whole depth of the mixed layer. The stratification index was $O(1000J/m^2)$ in the Bering Sea basin and the southern Bering Sea shelf including BL01-BL13, as shown in Figure 10 (a) and (b). In the northeastern Bering Sea shelf, due to the high-salinity of the Anadyr Water, the SI was significantly larger (Figure 10 (c)). In the northwestern Bering Sea shelf and the Chukchi Sea, the SI was significantly smaller (Figure 10 (c), (d), (e), and (f)). The SI showed a trend of decrease northward and was dominated by the salinity. The contribution of the temperature to stratification was too weak to be ignored. This was consistent with the research of Johnson et al. (2012), which showed that the seasonal variation of the mixed layer in the Arctic was dominated by salinity. Therefore, it was reasonable to assume that the characteristics of the mixed layer are related to the low-salinity water generated from the melting of sea ice in the Chukchi Sea and the northern Bering Sea shelf in the summer of 2019.*

[Figure]

Figure 22 **Figure 10.** *(a)~(f) The left axis represented the stratification index. Red was the proportion of stratification due to temperature. Green was the proportion due to salinity. The right axis represented the percentage of the contribution of the temperature. The blue dashed line represented the proportion of the contribution of the temperature to the stratification at different stations. (g)~(l) The mean Turner Angle within the mixed layer.*

l274-278: Besides a stratification index you may also consider the Turner angle (e.g, Clement et al, 2020, or references therein).

Reply:
Thank you for your suggestion. The Turner angle was showed in Figure 23 and analyzed in the revised manuscript, showed in the following:

*The mean Turner angle within the mixed layer was $-45° < Tu < 45°$, which meant that the mixed layer was stable on the whole. Half of the stations in the Bering Sea was larger than $0°$, and that meant that temperature played important role in the stable layer. The contrast along the BS section showed that temperature dominant the stable state in the Anadyr Water while salinity dominant the stable state in the Alaska Coastal Water. All the station except the R02 showed Turner angles smaller than $0°$, which meant that the salinity dominant the stable state in the Chukchi Sea.*

[Figure]

Figure 23 *Figure 10 (a)~(f) The left axis represented the stratification index. Red was the proportion of stratification due to temperature. Green was the proportion due to salinity. The right axis represented the percentage of the contribution of the temperature. The blue dashed line represented the proportion of the contribution of the temperature to the stratification at different stations. (g)~(l) The mean Turner Angle within the mixed layer.*

l290-322: This is a mere description of your figures. Please present new, original, scientific results: is there something unexpected in the temperature and salinity in 2019 compared with the databases and the previously published literature?

Reply:
Reply you for your comments. I analyzed the observational dataset from previous Chinese National Arctic Research Expeditions and WOA as you suggested in the overall comments. The mixed layer was shallower in the summer of 2019 in the Bering Sea and Chukchi Sea than those in previous years and climatological fields from WOA. And this was accompanied by the warming of the mixed layer. And this results were added in the Section 4 Discussion in the revised manuscript. The results were showed in the following as well for your convenience:

*4.5 The inter-annual variation*

*To explore the inter-annual variation of the MLD in the Bering Sea and Chukchi Sea, the observations along the BL section and R section from the World Ocean Atlas 2018 (WOA2018) and previous Chinese National Arctic Research Expeditions were compared.*

[Figure]

Figure 24 *Figure 15 The inter-annual variation of the MLD, temperature, salinity, and density of the mixed layer from the Chinese National Arctic Research Expeditions and the climatological MLD from WOA along the BL section in the Bering Sea.*

*The MLD in 2019 was obviously shallower and the temperature of the mixed layer was higher than those in the other five years along the BL section in the Bering Sea (**Figure 15 (a) and (b)**). This shallower MLD was accompanied by the warming of the surface layer (**Figure 6 (c) and (d)**) and cold intermediate layer (CIL) (**Figure 16 (a) ~ (f)**). The minimum temperature of the CIL water mass in the BSb showed a trend of increase: it was 0.54℃, 0.94℃, 0.82℃, 0.69℃, 1.99℃, and 2.50℃ for the year of 1999, 2003, 2010, 2012, 2014, and 2019, respectively (**Figure 16 (a)~(f)**). The warming of the CIL may be related to the air temperature warming in the previous winter and the processes in the north pacific (Overland et al., 2012).*

[Figure]

Figure 25 *Figure 16 The temperature ((a)~(f)) and salinity ((g)~(l)) profiles along the BL section in the Bering Sea from the Chinese National Arctic Research Expeditions and WOA. These expeditions were all carried out in summer.*

*The MLD in 2019 was shallower than those in 1999, 2003, 2010, 2012, 2014, and 2017 along the R section in the Chukchi Sea (**Figure 17** (a)). And this was accompanied by the warming of the mixed layer (**Figure 17** (b), **Figure 8** (a) and **Figure 18**). This surface warming was related with to the regional air-sea heat flux and the Arctic amplification (Danielson et al., 2020). Chronologically, the salinity and density was*

*consistent with the WOA climatological fields (**Figure 17 (c) and (d)**), while the MLD was shallower and the temperature was higher than the WOA climatological fields in the summer of 2019 (**Figure 17 (a) and (b)**). But salinity dominated the spatial fluctuation of the density for most of the year (**Figure 17 (c) and (d)**). It should be noticed that the salinity of the water in the BSs was larger than the climatology (**Figure 6 (d) and (f), Figure 15 (c)**) while it was not so in the CSs (**Figure 17 (c)**). This may be linked to the increasing net glacial ablation in the Gulf of Alaska watershed (Danielson et al., 2020).*

[Figure]

Figure 26 *Figure 17 The inter-annual variation of the MLD, temperature, salinity, and density of the mixed layer from the Chinese National Arctic Research Expeditions and the climatological MLD from WOA along the R section in the Chukchi Sea.*

[Figure]

Figure 27 *Figure 18 The temperature ((a)~(f)) and salinity ((g)~(l)) profiles along the R section in the Chukchi Sea from the Chinese National Arctic Research Expeditions and WOA. These expeditions were all carried out in summer.*

l327 " the BL section was representative" representative of what?

Reply:
Thank you. I cleared my expression as following:
*The BL section was representative main circulation and water masses during the expedition period in both the Bering Sea basin and shelf.*

l370, Figure 5: the labels on the graphs could be more readable.

Reply:
Thank you. I modified the labels Figure 5 and supplemented the lines representing the MLD as well. The new Figure 5 in the revised manuscript was as following:

[Figure]

Figure 28 *Figure 5 The upper panels and the lower panels represented the temperature and salinity profiles, respectively. The left (a, d), middle (b, e), and right (c, f) column represented the section of BL, BR, and BS, respectively. The blue solid line represented the MLDd. The magenta dashed line represented the MLDt.*

l374: Figure 6 : You don't need to show the sea surface temperature and salinity, unless there is something new. Does your measure SST compare will with satellite SST? Does your SSS compare with the climatology (say, World Ocean Atlas) for the month of the cruise? If the year 2019 is special, how and

why?

Reply:

Thank you for your suggestions. I compared the in situ measure SST with the satellite SST, and it was consistent with the satellite SST (the difference within 1℃, Figure 29 (c) and (e)). The measure SST was warmer (reached 2.7 ℃) than the climatology (WOA, Figure 29 (c) and (e))). The measure SSS was consistent with the climatology (WOA) in the Bering Sea basin but larger than the climatology in the Bering Sea shelf (Figure 29 (d) and (f)). The detailed discussion was showed in the revised manuscript:

*The MLD in 2019 was obviously shallower and the temperature of the mixed layer was higher than those in the other five years along the BL section in the Bering Sea (Figure 16 (a) and (b)). This shallower MLD was accompanied by the warming of the surface layer (Figure 6 (c) and (d)) and cold intermediate layer (CIL) (Figure 17 (a) ~ (f)).*

*It should be noticed that the salinity of the water in the Bering Sea shelf was larger than the climatology (Figure 6 (d) and (f), Figure 16 (c)) while it was not so in the Chukchi Sea shelf (Figure 18 (c)). This may be linked to the increasing net glacial ablation in the Gulf of Alaska watershed (Danielson et al., 2020).*

[Figure]

Figure 29 *Figure 6 The sea surface temperature (a) and salinity (b) in the Bering Sea.*

l388: what are the units for the rate?

Reply:

The rate equals $\Delta MLD(m)$ divided by distance (m). So, the rate was dimensionless because both the units of the MLD and distance was meter. I supplemented this explanation in the revised manuscript as well.

l427-428, "The MLD in the Bering Sea shelf fluctuated with the topography": Where is this demonstrated? Is there a figure to show the relationship between MLD and bathymetry?

Reply:
Thank you. I supplemented the MLD in Figure 28 (Figure 5 in the revised manuscript), and bathymetry was showed as well. The related stations and figures were listed in the revised manuscript:
*The MLD at the stations BL07~BL14 and BR01~BR09 in the Bering Sea shelf fluctuated with the topography (**Figure 5** (a) and (b)).*

l429: How can bottom friction constrain MLD? What is the physical process, what is the evidence?

Reply:
Thank you for your insightful comment. I deleted my assumption in the revised manuscript. I need to do sensitivity experiment by numerical modeling to explore the evidence and the physical process in my following research.

l430-433: This is irrelevant. Here you link the ML depth to the position of isotherms, but if there is a dynamic link, it has to be between the MLD and the seasonal pycnocline (the underlying stratification).

Reply:
Thank you for your comment. I deleted this irrelevant part in the revised manuscript. I will do sensitivity experiment by numerical modeling to explore the dynamic link in my following research.

l434-441: This may be interesting, but it needs to be discussed in relation with the litterature. What have you found that is new?

Reply:
Thank you. Previous researches on the Anadyr Water and the Alaska Coastal Water focus on the temperature and salinity and its' distribution: In the northwestern Bering Sea shelf, there was a cold and salt water mass called the Anadyr Water (AW hereafter) (Wang et al., 2020; Liu et al., 2016). The Alaska Coastal Water (ACW hereafter) was located on the northeastern Bering Sea shelf with the feature of high temperature and low salinity (Wang et al., 2020; Liu et al., 2016). In this study, their impact on mixed layer was discussed:
*Due to the significant difference in density between the AW and the ACW, advection occurred and the seawater was stratified in the transition zone. As a result, the MLD in the transition zone was shallower*

*than that in the northeastern and northwestern BSs ( Figure 7 (c)).*

l442-444: Is this a consistent with Peralta-Ferriz & Woodgate, 2015? Or do you find something different?

l453-455: same question as above. Is this a consistent with Peralta-Ferriz & Woodgate, 2015? Or do you find something different?

Reply: to both l442-444 and l453-455

Thank you for your comments. The research by Peralta-Ferriz &Woodgate, 2015 was vast and numerous, and it was very great. It inspired me. The research by Peralta-Ferriz &Woodgate, 2015 focus the spatial distribution the MLD in Chukchi Sea in summer **in a wider space range**: *Summer MLDs, everywhere shallower than winter MLDs, show a smaller but spatially similar east to west decrease (Eurasian Basin ~22 m, Makarov Basin ~16 m, Canada Basin and Southern Beaufort Sea ~9 m), with the Chukchi (~12 m) being regionally perhaps slightly deeper. The Barents Sea, although giving by far the deepest winter MLDs (~168 m), has summer MLDs that are in general a little shallower (~18 m) than the adjacent Eurasian Basin (~22 m).*

It neither mentioned "*northward increase of the MLD in the Chukchi Sea*", nor discussed "*the larger MLD at R05 and R07 stations might be related to the ACW appearing within the range of 68.5 - 70.5°N on the bottom.*"

Overall, this study was more regional and specific to the Chukchi Sea. The mean MLD in the Chukchi Sea was 10 m, smaller than that of Peralta-Ferriz &Woodgate, 2015. This MLD shoaling was discussed in the section 4.5 The inter-annual variation, which was already listed above responding to comments on l290-322.

l447-451: parallel to, perpendicular to: wrong grammar.

Reply:

I corrected them as following in my revised manuscript:

*Although the MLD increased in the Chukchi Sea slope as that in the Bering Sea slope, there was a difference between them. It was remarkable that, from the ocean basin towards the continental shelf, the isotherm and isohaline tended to **be in parallel with** the continental slope in the Chukchi Sea, while they tended to **be perpendicular to** the continental slope in the Bering Sea.*

l467-l490, Stratification: can you focus on what it new?

Reply:

Thank you. The Stratification Index was recalculated within the mixed layer in the revised manuscript other than the 60 m water column. And the role of the temperature and salinity in the MLD was discussed by Stratification Index and Turner angle:

*The SI covered the whole depth of the mixed layer. The stratification index was O( ) in the Bering Sea basin and the southern Bering Sea shelf including BL01-BL13, as shown in Figure 10 (a) and (b). In the northeastern Bering Sea shelf, due to the high-salinity of the Anadyr Water, the SI was significantly larger (Figure 10 (c)). In the northwestern Bering Sea shelf and the Chukchi Sea, the SI was significantly smaller (Figure 10 (c), (d), (e), and (f)). The SI showed a trend of decrease northward and was dominated*

*by the salinity. The contribution of the temperature to stratification was too weak to be ignored. This was consistent with the research of Johnson et al. (2012), which showed that the seasonal variation of the mixed layer in the Arctic was dominated by salinity. Therefore, it was reasonable to assume that the characteristics of the mixed layer are related to the low-salinity water generated from the melting of sea ice in the Chukchi Sea and the northern Bering Sea shelf in the summer of 2019.*

[Figure]

Figure 30 *Figure 10 (a)~(f) The left axis represented the stratification index. Red was the proportion of*

*stratification due to temperature. Green was the proportion due to salinity. The right axis represented the percentage of the contribution of the temperature. The blue dashed line represented the proportion of the contribution of the temperature to the stratification at different stations. (g)~(l) The mean Turner Angle within the mixed layer.*

*The mean Turner angle within the mixed layer was $-45° < Tu < 45°$, which meant that the mixed layer was stable on the whole. Half of the stations in the Bering Sea was larger than $0°$, and that meant that temperature played important role in the stable layer. The contrast along the BS section showed that temperature dominant the stable state in the Anadyr Water while salinity dominant the stable state in the Alaska Coastal Water. All the station except the R02 showed Turner angles smaller than $0°$, which meant that the salinity dominant the stable state in the Chukchi Sea.*

l471: "temperature interpreted": awkward style.

Reply:
Thank you. As the paragraph was rewrite as replied to above comment, this awkward style was deleted and the whole revised manuscript was corrected as well. And I corrected this awkward style throughout the manuscript.

486, figure 10: I don't understand the figure, I don't understand the axes. I suppose that if temperature explains x%, then salinity explains (100-x)% of the stratification, isn't it? In that case, information about one of the two is sufficient to deduce the other. What other useful information is there in the figure?

Reply:
Thank you for your comments. The revised figure 10 was showed as Figure 30 in this reply letter. It should be noticed that there were two y-axes for one x-axis. And the left y-axis represented the stratification index (red for temperature and blue for salinity) and the right y-axis represented the percentage of temperature. The useful information in the figure was that the salinity dominated the stratification, so the spatial distribution of the MLD was related to the processes influencing the salinity.

l507, figure 11: the arrows are unreadable. What are the red contours? What is the 16-days period over which you have averaged the data?

Reply:
Thank you. The figure 11 was revised to make the arrows readable. And these explanations were supplemented in the revised manuscript as following (**But it was deleted in the revised manuscript as you suggested in the next comment**):

[Figure]

Figure 31 *Figure 11. The 16-day (the period during which the Expedition was carried out) averaged absolute dynamic topography and the surface geostrophic flow in the Bering Sea and the Chukchi Sea from satellite altimeter. The red, yellow, and black solid line represented the 200 m, 2000 m, and 3000 m isobaths, respectively.*

l511, figure 12: the arrows are much easier to read in this figure compared with figure 11. Figure 11 seems redundant.

Reply:
Thank you. I checked the manuscript and modified figures to make them easier to read.

l492-493: Regarding the deepening of the MLD in the Bering Sea slope, is there an influence of tidal mixing? Are internal tides generated along the slope? Tidal influences may be larger than eddy influience there.

Reply:
Thank you for your insightful comment. I need more measure data and sensitivity experiment by numerical modeling to explore the influence of tidal mixing on the deepening of the MLD in the Bering Sea slope in my following research.

l497-499: "probably related to the eddies"... Is it related or not? You can reach a stronger conclusion, based on the data available. It is important do discuss the expected behavior of MLD in cyclones. vs. anticyclones (Gaube et al, 2019 and references therein). Do your measurements confirm or contradict the litterature?

Reply:
Thank you for your comments. I reached a stronger conclusion in the revised manuscript, and this

confirm the conclusion that anticyclones deepen the MLD in the research of Gaube et al., 2019:

*The large MLD at BL01 in the northern continental slope of the Aleutian Islands was related to the anticyclonic eddies along the Aleutian Islands (Figure 11). And this coincides with the conclusion that anticyclones deepen the MLD in the research of Gaube et al., 2019. The MLD at BL01 was 30 m, significantly larger than that at BL02, which was 19 m (Figure 7 (a)). The upper ocean current velocity at BL01was about 0.2m/s, while it was measured less than 0.1m/s in the BSb according to the ADCP observations. The spiral of the current became irregular at the base of the mixed layer at BL01 (Figure 11 (c)).*

l520 "In summer, the Aleutian low moved northward": why the use of the past tense? Do you mean the summer of 2019 in particular, or do you mean that 2019 was like every summer?

Reply:

Thank you. I mean that 2019 was like every summer. I should not use the past tense, and I corrected it in the revised manuscript:

In summer, the Aleutian Low moves northward and the south wind prevails.

l522-525: I don't understand why you are trying to correlate zonal and meridional winds, what are the time or space scales you compute your correlations over, and what you mean by "behaved well".

Reply:

Thank you. Sorry for my ambiguous expression. I correlated the measured zonal winds and the Cross-Calibrated Multi-Platform (CCMP) zonal winds. Meanwhile, I correlated the measured meridional winds and the Cross-Calibrated Multi-Platform (CCMP) meridional winds. I computed the correlations over the period from 24 Aug. to 6 Sep., during which the measurement of temperature and salinity was carried out. I mean that CCMP wind coincide with the measured wind: The correlation coefficients of the zonal wind between the CCMP wind and the measured wind by the ship were 0.92. The correlation coefficients of the meridional wind between the CCMP wind and the measured wind by the ship were 0.91. And these corrected expression was supplemented in the revised manuscript:

*The speed of the ship estimated by GPS was used to calculate wind speed. The sampling interval was 1 minute. The wind observed by the shipborne automatic meteorological station were used to evaluate the Version 2 Cross-Calibrated Multi-Platform (CCMP) Wind Vector Analysis Product over the period from 24 Aug. to 6 Sep.. The wind speed bias, wind speed root-mean-square error (RMSE hereafter), wind direction RMSE of the CCMP wind product was 1.29m, 2.37m, and $27.46°$, respectively. The correlation coefficients of the zonal wind between the CCMP wind and the measured wind by the ship were 0.92. The correlation coefficients of the meridional wind between the CCMP wind and the measured wind by the ship were 0.91. The mean difference of the zonal wind between the CCMP wind and the wind measured by the ship was 0.51 m/s. And the mean difference of the meridional wind between the CCMP wind and the wind measured by the ship was 0.29 m/s. That meant the CCMP wind product behaved well in the target region.*

l526-534: The correlation between wind and MLD is not convincing. The mixed layer deepening due to strong wind is a process that takes time (at least one inertial period), and it is very sensitive to wind bursts at high frequency. What is the frequency of your wind product? Maybe you should try to correlate each

point observation with the wind rms amplitude integrated over the previous half day or day. What is the spatial variability of the wind? Does the amplitude of the wind speed vary significantly from one hydrographic section to the next?

Reply:
Thank you for your comments. The vessel measured the wind along the way per minute, and the CCMP wind is a 6-hourly ocean vector wind analysis product. As you suggested, I correlated each point observation with the CCMP wind Root Mean Square (RMS) amplitude integrated over the previous day. On this occasion, the wind and MLD showed better correlation, and the correlation coefficient was changed from 0.6 to 0.63. The amplitude of the wind speed varied significantly from one hydrographic section to the next, and this could be seen from the scatter plot of MLD and wind speed (Figure 32).

[Figure]

Figure 32 *Figure 13. Scatter plot of wind speed and MLD of all the stations. The red solid line was the regression line between the wind speed and the MLDD in the BL (except BL01), BR, BT, and M stations. The blue solid line was the regression line between the wind speed and the MLDD of all the stations.*

l536: Figure 13. Explain in the legend what the two regression lines are. Why are there data points along the blue line? Why are there points with zero MLD?

Reply:
Thank you. I supplemented what the two regression lines are in the legend in Figure 13 (Figure 32 in this reply letter): The red solid line was the regression line between the wind speed and the MLD in the BL (except BL01), BR, BT, and M stations. The blue solid line was the regression line between the wind speed and the MLD of all the stations. The BL01 was excluded because the deepening of MLD was attributed to the eddies, while the MLD at R was characterized by the front formed by the high-density Anadyr Water and the low-density Alaska Coastal Water.
As for the data points along the blue line, I plotted this figure using the dash-dot line style in MATLAB. I changed this misleading line style to solid line in the revised manuscript (Figure 32 in this reply letter).

I correlated the MLD from temperature instead of MLD from density in the previous manuscript, which resulted in the existence of points with zero MLD. I corrected this and correlated the MLD from density and wind speed in the revised manuscript (Figure 32 in this reply letter) .

l542 "was shown": why use the past tense here?

Reply:

Thank you. I corrected it into present tense now:

*The average buoyancy flux caused by sea surface net heat flux and freshwater flux from July 1 to Sept. 8 is showed in Figure 13.*

l538-547: I am not sure any of the correlations you compute are significant. Please compute the significance of each correlation and eliminate all correlations that are not significant from the discussion and from the figures.

Reply:

Thank you for your comments. The significance of the correlation between the MLD and the combined effect of the buoyancy flux and the momentum was 0.046 (P value), smaller than 5% and indicating that the correlation was convincing. All correlations that are not significant were eliminated from the discussion:

*The significance of the correlation between the MLD and the combined effect of the buoyancy flux and the momentum was 0.046 (P value), smaller than 5% and indicating that the correlation was convincing. Under the combined effect of buoyancy flux and momentum flux, the MLD could reach a regional extremum, such as BL14, BR00, BR11, BT12, BT25, BT26, M11, R01, R05, R08, R11 in Figure 13. This result of multiple linear regression had a correlation coefficient of 0.41 with the measured MLD.*